# Waning habitats due to climate change: the effects of changes in streamflow and temperature at the rear edge of the distribution of a cold-water fish

José M. Santiago[1], Rafael Muñoz-Mas[2], Joaquín Solana[1], Diego García de Jalón[1], Carlos Alonso[1], Francisco Martínez-Capel[2], Javier Pórtoles[3], Robert Monjo[3], Jaime Ribalaygua[3]

[1]Laboratorio de Hidrobiología, ETSI Montes, Forestal y Medio Natural, Universidad Politécnica de Madrid, Camino de las Moreras s/n, Madrid 28040, Spain

[2]Institut d'Investigació per a la Gestió Integrada de Zones Costaneres (IGIC), Universitat Politècnica de València. C/ Paranimf 1, Grau de Gandia, València 46730, Spain

[3]Fundación para la Investigación del Clima, C/ Tremps 11. Madrid 28040, Spain

*Correspondence to*: José M. Santiago (jmsant@picos.com)

**Abstract.** Climate changes affect aquatic ecosystems by altering temperatures and precipitation patterns, and the rear edges of the distributions of cold-water species are especially sensitive to these effects. The main goal of this study was to predict in detail how changes in air temperature and precipitation will affect streamflow, the thermal habitat of a cold-water fish (the brown trout, *Salmo trutta*), and the synergistic relationships among these variables at the rear edge of the natural distribution of brown trout. Thirty-one sites in 14 mountain rivers and streams were studied in central Spain. Models of streamflow were built for several of these sites using M5 model trees, and a non-linear regression method was used to estimate stream temperatures. Nine global climate models simulations for the RCP4.5 and RCP8.5 (Representative Concentration Pathway) scenarios were downscaled to the local level. Significant reductions in streamflow were predicted to occur in all of the basins (max. -49 %) by the year 2099, and seasonal differences were noted between the basins. The stream temperature models showed relationships between the model parameters, geology and hydrologic responses. Temperature was sensitive to streamflow in one set of streams, and summer reductions in streamflow contributed to additional stream temperature increases (max. 3.6°C), although the sites that are most dependent on deep aquifers will likely resist warming to a greater degree. The predicted increases in water temperatures were as high as 4.0°C. Temperature and streamflow changes will cause a shift in the rear edge of the distribution of this species. However, geology will affect the extent of this shift. Approaches like the one used herein have proven to be useful in planning the prevention and mitigation of the negative effects of climate change by differentiating areas based on the risk level and viability of fish populations.

## 1 Introduction

Water temperatures are a primary influence on the physical, chemical and biological processes in rivers and streams (Caissie, 2006; Webb *et al.*, 2008) and, subsequently, the organisms that live completely or partially in the water. Temperature is a

major feature of the ecological niche of poikilothermic species (*e.g.*, Magnuson and Destasio, 1997; Angilletta, 2009) and a key factor in energy balance of fish. It affects the rate of food intake, metabolic rate and growth performance (Forseth *et al.*, 2009; Elliott and Elliott, 2010; Elliott and Allonby, 2013). It is also involved in many other physiological functions, such as blood function and reproductive maturation (Jeffries *et al.*, 2012), reproductive timing (Warren *et al.*, 2012), gametogenesis (Lahnsteiner and Leitner, 2013), cardiac function (Vornanen *et al.*, 2014), gene expression (White *et al.*, 2012; Meshcheryakova *et al.*, 2016), ecological relationships (Hein *et al.*, 2013; Fey and Herren, 2014), and fish behaviour (Colchen *et al.*, 2016).

Natural patterns of water temperature and streamflow are profoundly linked with climatic variables (Caissie, 2006; Webb *et al.*, 2008). Therefore, stream temperature is strongly correlated with air temperature (Mohseni and Stefan, 1999), whereas streamflow has a complex relationship with precipitation (McCuen, 1998; Gordon *et al.*, 2004). In addition, atmospheric temperature influences the type of precipitation (rain or snow) that occurs and the occurrence of snowmelt; conversely, river discharge is also a main explanatory factor of water temperature for some river systems (Neumann *et al.*, 2003; van Vliet *et al.*, 2011). Furthermore, geology affects surface water temperatures by means of groundwater discharge (Caissie, 2006, Loinaz *et al.*, 2013), influenced by the aquifer depth (shallow or deep) and the water's residence time (Kurylyk *et al.*, 2013, Snyder *et al.*, 2015).

Climate change is already affecting aquatic ecosystems by altering water temperatures and precipitation patterns. Stream temperature increases have been documented over the last several decades over the whole globe, such as in Europe (*e.g.*, Orr *et al.*, 2015, documented a mean increase in stream temperature of 0.03°C per year in England and Wales), Asia (*e.g.*, Chen *et al.*, 2016, documented a mean increase in stream temperature of 0.029-0.046°C per year in the Yongan River, Eastern China), America (*e.g.*, Kaushal *et al.*, 2010, documented mean increases in stream temperature of 0.009–0.077°C per year) and Australia (*e.g.*, Chessman, 2009, documented mean increases in stream temperature of 0.12°C per year between macroinvertebrate sampling campaigns). Abundant information is also available regarding the impact of recent climate changes on streamflow regimes worldwide (*e.g.*, Luce and Holden 2009; Leppi *et al.*, 2012) and, more specifically, in the Iberian Peninsula (*e.g.*, Ceballos-Barbancho *et al.*, 2007; Lorenzo-Lacruz e*t al.*, 2012; Morán-Tejeda *et al.*, 2014). However, detailed predictions are uncommon (*e.g.*, Thodsen, 2007). The predictions of the Intergovernmental Panel on Climate Change (IPCC, 2013) suggest that these alterations will continue throughout the XXI century, and they will have consequences for the distribution of freshwater fish (*e.g.*, Comte *et al.*, 2013; Ruiz-Navarro *et al.*, 2016). These changes may have an especially strong effect on cold-water fish, which have been shown to be very sensitive to climate warming (Williams *et al.*, 2015; Santiago *et al.*, 2016). For example, among salmonids, DeWeber and Wagner, (2015) found stream temperature to be the most important determinant of the probability of occurrence of brook trout, *Salvelinus fontinalis* (Mitchill, 1814).

The rear edge populations (*sensu* Hampe and Petit, 2005: "*populations residing at the current low-latitude margins of species' distribution ranges* […]") of a cold-water species are especially sensitive to changes in water temperature, in addition to reductions in the available habitable volume (*i.e.*, streamflow). The rear edge is the eroding margin of the range where lineages mix, the genetic drift and local adaptations increase, and droughts put populations under stress. The impact of water

temperatures on the distribution of salmonid fish is well documented (*e.g.*, Beer and Anderson, 2013; Eby *et al.*, 2014); however, the combined effects of rising stream temperatures and reductions in streamflow remain relatively unexamined; with some exceptions (*e.g.*, Wenger *et al.*, 2011, Muñoz-Mas *et al.*, 2016). Jonsson and Jonsson (2009) predicted that the expected effects of climate change on water temperatures and streamflow will have implications for the migration, ontogeny, growth
and life-history traits of Atlantic salmon, *Salmo salar* Linnaeus, 1758, and brown trout, *Salmo trutta* Linnaeus, 1758. Thus, investigation of these habitat variables in the context of several climate scenarios should help scientists to assess the magnitude of these changes on the suitable range and life history of these species.

The objective of this study is to predict how and to what extent the availability of suitable habitat for the brown trout, a sensitive cold-water species, will change within its current natural distribution under the new climate scenarios through a study of
changes in streamflow and temperature and their interactions. Specifically, in this paper, we (i) assessed the effects of both streamflow and geology on stream temperature; (ii) predicted the changes in streamflow and stream temperature implied by the climate change scenarios used in the 5$^{th}$ Assessment Report (AR5) of the IPCC; and (iii) assessed the expected effects of these changes on trout habitat aptitude. To this end, hydrologic simulations with M5 model trees coupled with non-linear water temperature models at the daily time step were fed with high-resolution, downscaled versions of the air temperature and
precipitation fields predicted using the most recent climate change scenarios (IPCC, 2013). The effects of basin geology on the stream temperature models and on the estimated changes in thermal regimes were studied. Finally, the changes in the thermal habitat of trout were assessed by studying the violation of the tolerable temperature thresholds of the brown trout.

## 2. Materials and methods

The logical framework followed is summarized in Fig. 1. First, the daily global climate models output presented by the IPCC
were downscaled to the study area. Then, the obtained local climate models output were applied to generate simulations of streamflow and water temperature. The results are daily values that can be used for the assessment of fish habitat suitability and availability.

The procedure yielded results in the form of continuous time series, but they are presented for two time horizons: the year 2050 (H-2050) and the year 2099 (H-2099). The values for these horizons correspond to the average of the values of the
different variables for the decades 2041-2050 and 2090-2099, respectively.

### 2.1 Study sites

In total, 31 sites in 14 mountain rivers and streams inhabited by brown trout were chosen with the aim of encompassing a diverse array of geological and hydrological conditions in the centre of Spain (between the latitudes of 39°53' N and 41°21' N). Specifically, the investigated sites are located in the Tormes River and its tributaries, the Barbellido River, the Gredos
Gorge and the Aravalle River (in the Duero basin); the Cega River and the Pirón River (the Pirón River is a tributary of the Cega River in the larger Duero basin); the Lozoya River, the Tagus River, the Gallo River, and the Cabrillas River (all four of

which are in the Tagus basin); the Ebrón River and the Vallanca River (the Vallanca River is a tributary of the Ebrón River in the Turia basin); and the Palancia River and the Villahermosa River (Fig. 2). The major geological components that lithologically characterize the mountain sites in the Duero and Lozoya basins are igneous rocks; the altitudinally lower sites in the Duero basin are underlain by Cenozoic detrital material, and the eastern basins (Tagus, Gallo, Cabrillas, Ebrón, Vallanca, Palancia and Villahermosa) are underlain primarily by Mesozoic carbonates. The distribution of geological materials was retrieved from the Lithological Map of Spain (IGME, 2015) (Table 1).

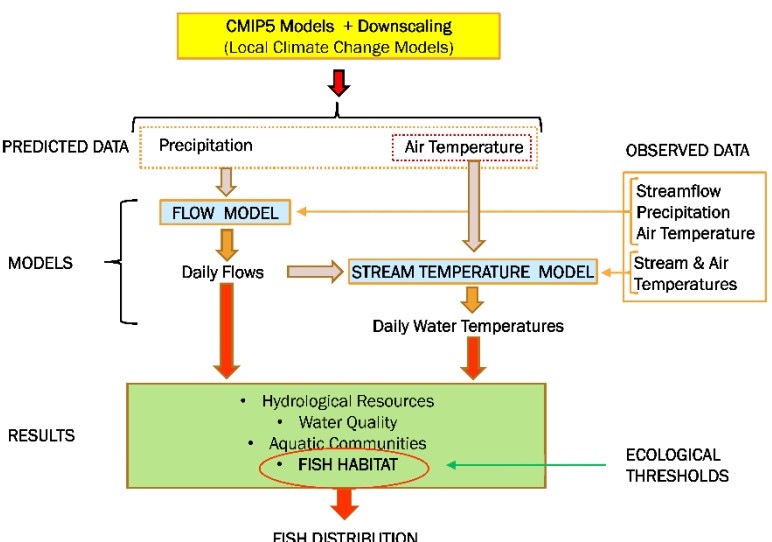

**Figure 1. Logical framework of the study.**

The land cover type is mainly pine forest in all of the studied basins (*Pinus sylvestris*, *P. nigra*, *P. pinea* and *P. pinaster*) (CORINE Land Cover 2006, European Environmental Agency, [2007]). Only the lower basins of the downstream sites on the Cega and Pirón rivers are mosaics of forest and croplands, whereas the uppermost sites within the Tormes River basin (Barbellido and Gredos Gorge) lie above the current tree line. Territorial planning does not consider significant changes in land-use at mid-century; objectively, changes are not expected after that time because a high percentage of the territory is protected. The studied reaches are not effectively regulated (only small weirs or natural obstacles exist). One large dam lies on the Pirón River (the Torrecaballeros Dam, which has a capacity of 0.32 hm$^3$ and a maximum depth of 26 m and lies at an altitude of 1390 m a.s.l.), but it does not significantly alter the temporal pattern of streamflow (Santiago *et al.*, 2013). In the Lozoya River, a large dam (the Pinilla Dam, which has a capacity of 38.1 hm$^3$ and a maximum depth of 30 m and lies at an altitude of 1060 m a.s.l.) exists that separates fish populations above and below the reservoir, although it lies downstream of the studied reach.

Hydrological data characterize the streamflow regimes as extreme winter/early spring (groups 13 and 14 in the classification of Haines *et al.*, [1988]). However, the hydrographs show a west-to-east smoothing gradient (Fig. 3). This smoothing is associated with the carbonate rocks, whereas greater seasonality is associated with the igneous and detrital geological materials.

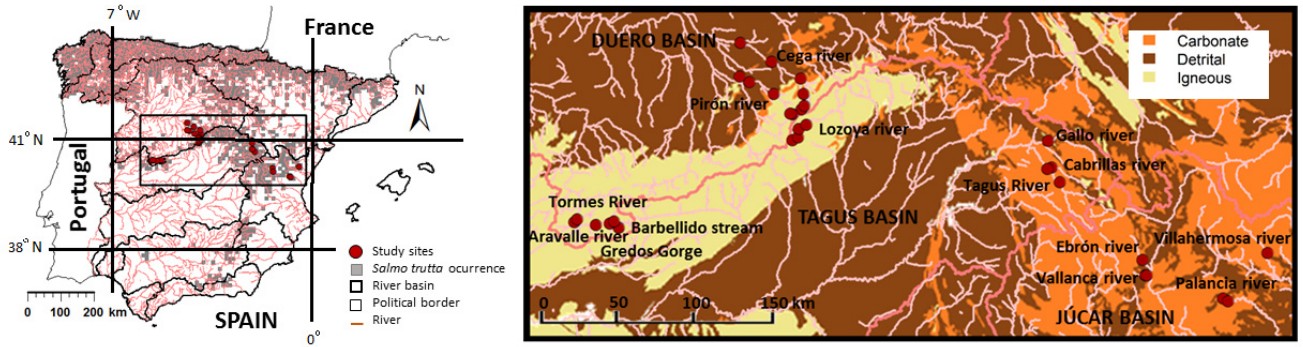

**Figure 2. River network and location of the study sites (water temperature data loggers), with details regarding lithology. The grid depicts the actual occurrence of brown trout in Spain.**

## 2.2 Data collection

At each study site, water temperatures were recorded every two hours throughout the year using 31 Hobo® Water Temperature

Pro v2 (Onset®) and Vemco® Minilog data loggers located at several sites along the studied rivers and streams (Table 1). Loggers were tested for malfunctions before being deployed, and they were placed in areas not exposed to direct sunshine (Stamp *et al*. 2014). Meteorological data were obtained from nine thermometric and 15 pluviometric stations of the Spanish Meteorological Agency (AEMET) network, and data from ten gauging stations (from the official network of the Water Administration) were obtained to model the streamflows. The AEMET-thermometric stations that lie closest to the stream

temperature monitoring sites and have at least 30 years of data between 1955 and the present were selected. The selected pluviometric stations were those located within the upstream river basin or near the corresponding gauging station (Table 2). The air temperature and precipitation data from AEMET were tested to assess their reliability by applying a homogeneity test. This test is based on a two-sample Kolmogorov–Smirnov test, and it marks years as possibly containing inhomogeneous data. In the second phase, the marked years are matched against the distribution of the entire series to determine if they contain true

inhomogeneities, searching for possible dissimilarities between the empirical distribution functions. Only reliable series were used. The locations of the stations did not change in the studied period.

## 2.3 Climate change modelling and downscaling

Data from nine global climate models associated with the 5th Coupled Model Intercomparison Project (CMIP5) were used, namely BCC-CSM1-1, CanESM2, CNRM-CM5, GFDL-ESM2 M, HADGEM2-CC, MIROC-ESM-CHEM, MPI-ESM-MR,

MRI-CGCM3, and NorESM1-M (Santiago *et al.*, 2016). These models provided daily data to simulate future climate changes

corresponding to the Representative Concentration Pathways RCP4.5 (a stable scenario) and RCP8.5 (a scenario including a pronounced increase in $CO_2$ concentrations) established in Taylor *et al.* (2009) and used in the AR5 of the IPCC (2013). An array of nine general climate models was used to avoid biases due to the particular assumptions and features of each particular model (Kurylyk *et al.*, 2013). Historical simulations of the XX century were used to control the quality of the procedure and to compare the magnitudes of the predicted changes.

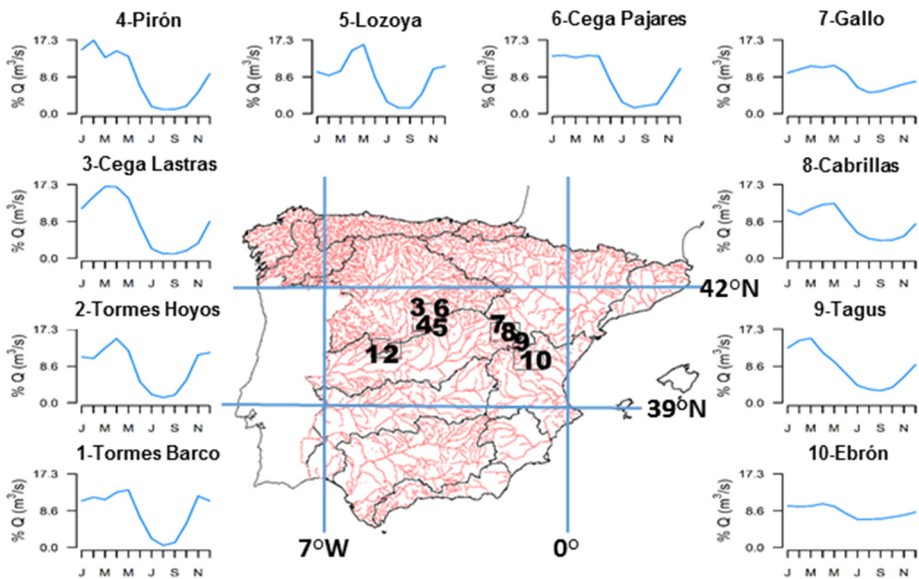

**Figure 3. River regime patterns for the different gauging stations. The flows are expressed as percentages of the mean annual flow, and the months are ordered from January to December.**

Pourmokhtarian *et al.* (2016) note the importance of the use of fine downscaling techniques. Thus, a two-step analogue statistical method (Ribalaygua *et al.*, 2013) was used to downscale the daily climatic data, specifically the maximum and minimum air temperatures and the precipitation for each station and for each day. For both air temperature and precipitation, the procedure begins with an analogue stratification (Zorita and von Storch, 1999) in which the *n* days most similar to each problem day to be downscaled are selected using four different meteorological large-scale fields as predictors, specifically (1) the speed and (2) direction of the geostrophic wind at 1000 hPa, as well as (3) the speed and (4) direction of the geostrophic wind at 500 hPa. In a second step, the temperature determination was obtained through multiple linear regression analysis using the selected *n* of the most analogous days. This was performed for the maximum and minimum air temperatures at each station and for each problem day. The linear regression uses forward and backward stepwise selections of the predictors to select only the relevant predictive variables for that particular case. For precipitation, a group of *m* problem days (the whole days of a month were used) were downscaled together, and the "preliminary precipitation quantity," or the average precipitation of the *n* most analogous days, was obtained for each problem day. Thus, the *m* problem days from the highest to the lowest 'preliminary precipitation amount' could be sorted. To assign the final amount of precipitation, each precipitation amount of

the $m \times n$ analogous days was taken. Then, those $m \times n$ amounts of precipitation were sorted, and then those amounts were clustered into $m$ groups. Every quantity was then assigned in order to the $m$ days previously sorted by the 'preliminary precipitation amount'. Further details of the methodology are described in Ribalaygua *et al.* (2013).

A systematic error is obtained when comparing the simulated data from the climate models with the observed data. Such errors are inherently associated with all downscaling methodologies and climate models, which usually introduce bias into their outputs. To eliminate this systematic error, the future climate projections were corrected according to a parametric quantile-quantile method (Monjo *et al.*, 2014), which was performed by comparing the observed and simulated empirical cumulative distribution functions (ECDF) and linking them using ECDFs obtained from the downscaled European Centre for Medium-Range Weather Forecasts ERA-40 reanalysis daily data (Uppala *et al.*, 2005).

As a result, for each climate change scenario, the daily maximum and minimum air temperatures (which were used to infer the mean air temperature) and precipitation were obtained for each climate model, and the whole dataset were used as inputs to simulate the runoff and water temperatures under these climate change scenarios.

## 2.4 Hydrological modelling

Although process-based physical models are considered the standard hydrological models, flexible data-driven machine learning techniques are gaining popularity because they can be based solely on precipitation and temperature (Shortridge *et al.*, 2016) and can be automatized to perform multiple simulations. Therefore, the prediction of the streamflows under the climate change scenarios was performed with data-driven hydrological models developed using the M5 algorithm (Quinlan, 1992). M5 has been shown to have skill in modelling daily streamflow (Solomatine and Dulal, 2003; Taghi Sattari *et al.*, 2013), including in studies involving climate change (Muñoz-Mas *et al.*, 2016), and it is sufficiently fast to deal proficiently with larger datasets (Quinlan, 2017)

Mathematically, M5 is a kind of decision tree that, instead of assigning a single value to each terminal node, assigns a multi-linear regression model (Quinlan, 1992). Therefore, the dataset is hierarchically divided in homogeneous parts and a multi-linear model is adjusted to every part (Hettiarachchi *et al.*, 2005). In this regard, each node, and the corresponding multi-linear regression model, is specialized in particular areas of the data set, such as peak flows or base flows, to name the extremes (*i.e.*, it is a piece-wise linear model with each part dedicated to a particular hydrologic condition) (Taghi Sattari *et al.*, 2013). Based on the multi-linear models at the terminal nodes, M5 allows extrapolation, in contrast with other machine learning techniques that have demonstrated little or no extrapolation ability (*e.g.*, random forest or multilayer perceptron) (Hettiarachchi *et al.*, 2005, Shortridge *et al.*, 2016).

The M5 hydrological models were developed in R (R Core Team, 2015) with the *Cubist* package (Kuhn *et al.*, 2014). One single M5 model tree was trained for each gauging station (ten models were produced in total; Fig. 3 and Table 2), whereas the predictions were supported by the nearest observation (*i.e.*, neighbours=1) to avoid producing unreliable flows. Finally, M5 was allowed to determine the ultimate number of models (*i.e.*, nodes or areas) into which the dataset is eventually divided (see Kuhn *et al.*, 2014).

Following previous studies, the M5 hydrological models were trained by employing, the daily, monthly and quarterly data lags of historical precipitation and air temperature collected at meteorological stations within or nearby the target river basins as input variables (Table 2) (Solomatine and Dulal, 2003; Taghi Sattari *et al.*, 2013; Muñoz-Mas *et al.*, 2016). These three groups of variables were intended to reflect the causes of peak, normal and base flows. The study encompassed several rivers and

streams that may have different hydrologic behaviours; therefore, the starting set of input variables, which was afterwards subset, was larger than that used in other studies (Solomatine and Dulal, 2003; Taghi Sattari *et al.*, 2013; Muñoz-Mas *et al.*, 2016). The daily variables included the precipitation and air temperature from the current day to the 15th previous day (16 variables in total). The monthly variables were calculated using the moving average for the 12 previous months (12 variables in total), and the quarterly data were calculated from the moving average for the current month to the 24th previous month (8

variables in total). Consequently, the daily variables overlapped with the current month variable, and the first four quarterly variables overlapped with the monthly data. In the end, 72 variables were gathered, 36 each for air temperature and precipitation.

The whole set of input variables may be relevant for some river systems (Shortridge *et al.*, 2016), although it may cause M5 to overfit the data in others (Schoups *et al.*, 2008). Therefore, the ultimate variable subset was optimized following the forward

stepwise approach (Kittler, 1978). This approach relies on iteratively adding input variables (one at a time) while the performance on the test data set improves and stopping (*i.e.*, selecting a smaller subset of the input variables) as soon as the performance stagnates or degrades. However, the classical forward stepwise approach may cause consideration of unrelated variable sets (*i.e.*, disjoint precipitation and air temperature variable lags). To address such potential inconsistencies, the optimization began by testing the precipitation-related variables and only tested the air temperature variables for lags

coinciding with those precipitation-related variables that were already selected. No precautions were taken regarding correlations among inputs (Solomatine, personal communication), and the forward stepwise approach sought to maximize the Nash-Sutcliffe efficiency (NSE) index (which ranges from −∞ to 1; Nash and Sutcliffe [1970]) in a fivefold cross-validation (*i.e.*, for each combination of variables, five M5 model trees were trained on four parts and validated with the fifth part, which was held out) (Borra and Di Ciaccio, 2010, Bennett *et al.*, 2013). Finally, in order to account for the uncertainty of the models

(Bennett *et al.*, 2013), the variance of the NSEs obtained during the cross-validation was inspected; large intervals led to alternative data partitions. Following previous studies (Fukuda *et al.*, 2013; Muñoz-Mas *et al.*, 2016), once the optimal variables set for each gauging station was determined, ten M5 model trees (*i.e.*, one per gauging station) were developed using the corresponding subset of variables, and they were used to predict the streamflows under the climate change scenarios.

The daily data were analysed monthly and seasonally using the following statistics: minimum flow ($Q_{min}$), the 10th percentile

of flow ($Q_{10}$), the mean flow ($Q_{mean}$), and the maximum flow ($Q_{max}$). The annual runoff and days of zero flow were also examined.

To assess the significance of the streamflow trends throughout the century, Sen's slope was used (as implemented in the *Trend* package of R (Pohlert, 2016); p-value $\leq 0.05$) with horizons H-2050 and H-2099.

Finally, the variation of the patterns of the monthly mean streamflow was studied by means of the Ward Hierarchical Clustering implemented in the *cluster* R package (Maechler, 2013) on the basis of the rate of change of the normalized monthly mean streamflows in H-2050 and H-2099 and the RCP4.5 and RCP8.5 scenarios. Performance of the obtained cluster was quantified by using the agglomerative coefficient (a.c.). This is a measure of the clustering structure of the dataset, as expressed by Kaufman and Rousseeuw (2005) and its value ranges between 0 (maximum dissimilarity) and 1 (minimum dissimilarity).

## 2.5 Stream temperature modelling

Stream temperature ($Ts$) at each thermal sampling site was simulated from air temperature ($Ta$) by means of a modified version of the bounded non-linear regression model described by Mohseni *et al*. (1998). A previous modification (Term 1 in Eq. [1]; Santiago *et al*. [2016]) served to improve the behaviour of the former model, permitting it to be used for daily inputs. In this study, the effect of instream flow ($Q$) effect is incorporated. Thus, this model addresses daily mean stream temperature (DMST; $Ts$ in Eq. [1]) using the daily mean air temperature (DMAT, $Ta$ in Eq. [1]), the 1-day before variation of the daily mean air temperature ($\Delta Ta$ in eq.1), and the daily mean flow ($Q_{mean}$, $Q$ in eq.1) as predictors. DMST was used because it better reflects the average conditions that fish (particularly trout) will experience for an extended period of time (Santiago *et al.*, 2016), and it averages over daily fluctuations in the radiation and heat fluxes. The model is formulated as follows:

$$T_s = \underbrace{\mu + \frac{\alpha - \mu}{1 + e^{\gamma(\beta - T_a)}} + \lambda(\Delta T_a)}_{Term\ 1} + \underbrace{\frac{\omega}{1 + e^{\delta(\tau - Q)}}}_{Term\ 2} \qquad \text{Eq. (1)}$$

where $\mu$ is the minimum stream temperature (°C), $\alpha$ is the maximum stream temperature (°C), $\beta$ represents the air temperature at which the rate of change of the stream temperature with respect to the air temperature is a maximum (°C), $\gamma$ (°C$^{-1}$) is the value of the rate of change at $\beta$, and $\lambda$ is a coefficient (dimensionless) that represents the resistance of DMST to change with respect to the 1-day variation in DMAT ($\Delta Ta$). In the flow component (Term 2 in Eq. [1]), $\omega$ is the maximum observable variation in stream temperature due to the flow difference (given in °C), $\tau$ represents the flow value at which the rate of change of the stream temperature with respect to the flow is a maximum (m$^3$·s$^{-1}$), and $\delta$ (m$^{-3}$·s) is this maximum rate at $\tau$. Negative values of $\lambda$ are due to the resistance to stream temperature changes, and thus they must be subtracted from the expected temperature: the more resistant the stream is to temperature change, the closer $\lambda$ will be to zero. The less resistant the stream is to change, the more negative $\lambda$ is. The parameter $\mu$ was allowed to be less than zero in the modelling process, even though this is the freezing temperature. Thus, the function would truncate at the freezing point. The relationship between the thermal amplitude $\alpha$-$\mu$ and the indicator of thermal stability $\lambda$ was studied using the Pearson correlation.

A blockwise non-parametric bootstrap regression (Liu and Singh, 1992) was used to estimate the parameters of both the modified Mohseni models (with and without streamflow), and residual normality and non-autocorrelation were checked with the Shapiro test and Durbin-Watson test. Moreover, the seven-day lag PACF (partial autocorrelation function) was obtained. These calculations were performed using R. A 95 % confidence interval was calculated for each parameter. Performance was

 The Bayesian information criterion (BIC) and the Akaike information criterion (AIC) were used to test the eight-parameter models (Terms 1+2 of Eq. 1) against the five-parameter models (Term 1 of Eq. 1).

This model can be classified as semi-physically based mode. It has some advantages over machine learning methods, such as classification and regression trees (De'ath and Fabricius, 2000) or random forests (Breiman, 2001), because the model parameters imply a mechanistic interpretation of how process drivers act, yielding a higher transferability (Wenger and Olden, 2012). These features make of this model an advantageous option for our goals.

## 2.6 Effects of geology on stream temperature

Geology determines the residence time of deep groundwater in the aquifers underlying streams (Chilton, 1996), and residence times influence discharge temperatures. To explore the relationships between thermal regimes and geology, a stratified study of both the geology classes of the parameter values was completed by means of a t-test with the Bonferroni correction (p-value < 0.05). In the same sense, increments of the annual averages of the daily mean ($\Delta T_{mean}$), minimum ($\Delta T_{min}$) and maximum ($\Delta T_{max}$) stream temperatures were calculated and studied by lithological classes (Table 1)

The variation of the patterns of the monthly mean stream temperature was studied by means of cluster analysis of the temperature increases corresponding to H-2050 and H-2099 for the RCP4.5 and RCP8.5 scenarios (using Ward's hierarchical clustering as implemented in the *cluster* package of R; Maechler [2013]). As said above, agglomerative coefficient (a.c.) was used as a performance indicator.

## 2.7 Thermal habitat changes

Several tolerance temperatures and thermal niche limits have been described for brown trout (Table 3). The realized niche must reflect energetic efficiency: spending long periods above that threshold makes animals less efficient competitors, and their performance decreases critically (Magnuson *et al*., 1979; Verberk *et al*., 2016). Thus, we focused our study on the realized thermal niche. The elected threshold for this study was the occurrence of DMST values above 18.7°C for seven or more consecutive days, because it has proven to be the most realistic value to represent the realized thermal niche (Santiago *et al.*, 2016). The minimum period of seven consecutive days is usually the established time for determining thermal tolerance (Elliott and Elliott, 2010), and when this period is exceeded, the death risk (exclusion risk in our case) increases substantially. The chosen threshold was originally determined in one of the streams in this study (the Cega River).

Once DMST was modelled, the frequency of events of seven or more consecutive days above the threshold per year (Times Above the Threshold, TAT≥7), the total Days Above the Threshold per year (DAT), and the Maximum Consecutive Days Above the Threshold per year (MCDAT) were calculated for the whole period of 2015-2099.

To assess the general trend in thermal habitat alterations at the middle (H-2050) and the end of the century (H-2099), the TAT≥7, DAT and MCDAT were calculated at each sampling site for each climate change scenario and compared with current conditions

## 2.8 Longitudinal interpolation and extrapolation

The number of sampling sites and their distribution in the Cega, Pirón and Lozoya rivers (Fig. 2, Table 1) permit the longitudinal interpolation and extrapolation of the predicted water temperatures to study the relationships between the annual average DMST and altitude (strong correlations were detected between these quantities; $R^2$= 0.986, 0.985 and 0.881, respectively). A digital elevation model (DEM) with a resolution of 5 m made using LIDAR and obtained from the National Geographic Institute of the Spanish Government (IGN) was used to perform an altitudinal interpolation of the model parameters to determine the water temperature along the stream continuum to simulate the effects of the climate change scenarios and then to obtain the percentage of stream/river length that will be lost for trout. ArcGIS® 10.1 software (made by ESRI®) was used to manage the DEM.

All variables and acronyms are summarized in the Appendix 1. An overview of the uncertainty issue is given in Appendix 2.

## 3 Results

### 3.1 Climate change

Under the climate change scenarios, all the meteorological stations will experience noticeable temperature (DMAT) increases through the century. As might be expected, this trend is steeper for the RCP8.5 scenario, especially in summer, though it is also noticeable in winter to a lesser extent (annual trends are shown in Fig. 4; the seasonal results are shown by location in Fig. S1 to S24 [Supplementary Material 1]). The air temperature variations will run parallel one another in the two scenarios until mid-century, when the RCP8.5 scenario predicts a similar trend and the increases decrease under the RCP4.5 scenario; the annual change in temperatures for RCP4.5 fluctuates between 2°C and 2.5°C at mid-century and between 2.5°C and 3.5°C at the end of the century (3-4°C at mid-century and 3.5-4.5°C at the end of the century in summer) The annual change for RCP8.5 is between 2°C and 3°C at mid-century and between 5°C and 7°C at the end of the century (3.5-4.5°C at mid-century and 7-8°C at the end of the century in summer).

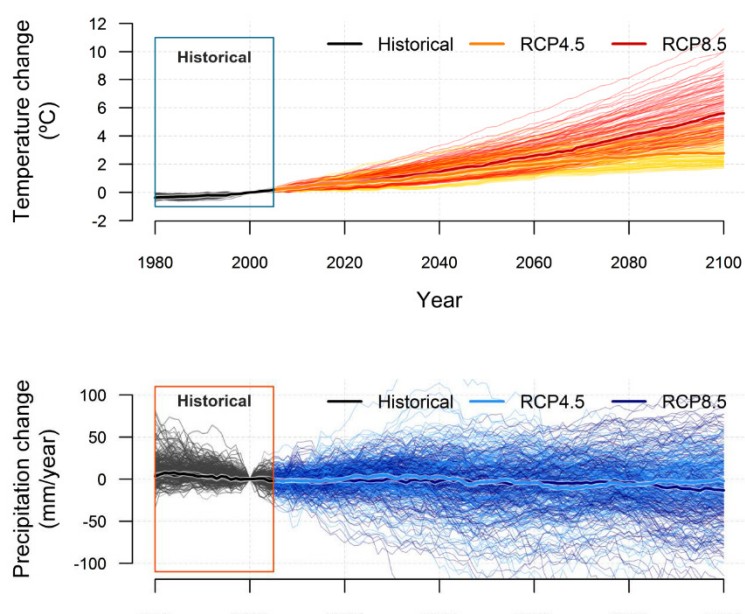

**Figure 4. Changes in mean air temperature and total annual precipitation related to climate change for the nine general climate models and the two climate change scenarios for the all the studied meteorological stations.**

The change in the annual precipitation (mm·day$^{-1}$) will fluctuate around zero (Fig. 4), although seasonal values will vary (Fig.
S1). RCP4.5 predicts a slight decrease (-7 %) by mid-century in total precipitation, which will return to current values by the end of the century. Conversely, RCP8.5 predicts stable precipitation up to mid-century and a slight decrease (-10 %) by the end of the century. The most important changes appear to occur in autumn. Daily mean air temperatures of the ensemble members for each meteorological station are shown in the Supplementary Material 2 (Dataset S1).

### 3.2 Hydrological regimes

In general, decreases in flow will occur throughout the century, but the degree of change will vary among the sites. Stations located in the western (Tormes) and eastern (Ebrón) extremes of the study area will experience an increase in flow by 2099 after decreasing in the mid-XXI century. Lozoya will suffer the most intense flow decreases, followed by Pirón and Cega-Lastras, Tagus and Gallo, and Cabrillas. These patterns of change in flow regimes are predicted to be linked to a West-to-East longitudinal gradient; climate change is expected to have less of an influence on discharge at the western stations and Ebrón
(in the far eastern portion of the study area).

The hydrological models performed well; all of them achieved NSE values ≥0.7 when a number of assorted combinations of variables were selected (Table S1 [Supplementary Material 3]). Fig. 5 shows plots of the monthly $Q_{mean}$ results of the

simulations for the RCP4.5 and RCP8.5 scenarios in H-2050 and H-2099. Daily mean streamflow estimated from the climate change model ensemble is given in the Supplementary Material 4 (Dataset S2).

### 3.2.1 RCP4.5 scenario

Statistically significant ($p<0.05$) shifts in the flow regime will be rare in H-2050 (Table 4, Fig. 5). In H-2099, these changes will be less pronounced, but significant changes become more frequent (Table 4, Fig. 5). Only two gauging stations (Lozoya and Tagus) exhibit significant reductions in annual discharge. By the end of the century (H-2099), annual discharge is expected to be significantly lower at seven gauging stations. Tagus basin will experience the greatest changes in annual discharge. Maximum, mean and minimum daily discharges ($Q_{mean}$ and $Q_{min}$), as well as the $Q_{10}$, will become much lower in Tagus River basin. Only Cega-Lastras and Pirón (Duero River basin) will suffer a significant increase in the number of zero-flow days.

### 3.2.2 RCP8.5 scenario

According to the predictions, the most significant changes in flow regimes will occur at the gauging stations of Cega-Lastras and Lozoya in H-2050 (Table 4, Fig. 5). In H-2099, most sites will experience strong flow reductions, even in seasons where seasonal increases in flow are predicted (*e.g.*, Ebrón and both stations in the Tormes River) (Table 4, Fig. 5). Significant annual runoff reductions in H-2050 will occur at five of the stations, increasing the occurrence of significant losses at nine out of the ten sites in H-2099 (*i.e.,* all stations except Ebrón). The most important decreases in every variable and throughout the century were predicted for the stations in the middle Cega basin and the Tagus basin. A significant increase in the number of days with no flow was predicted for Cega-Lastras, Pirón and Gallo.

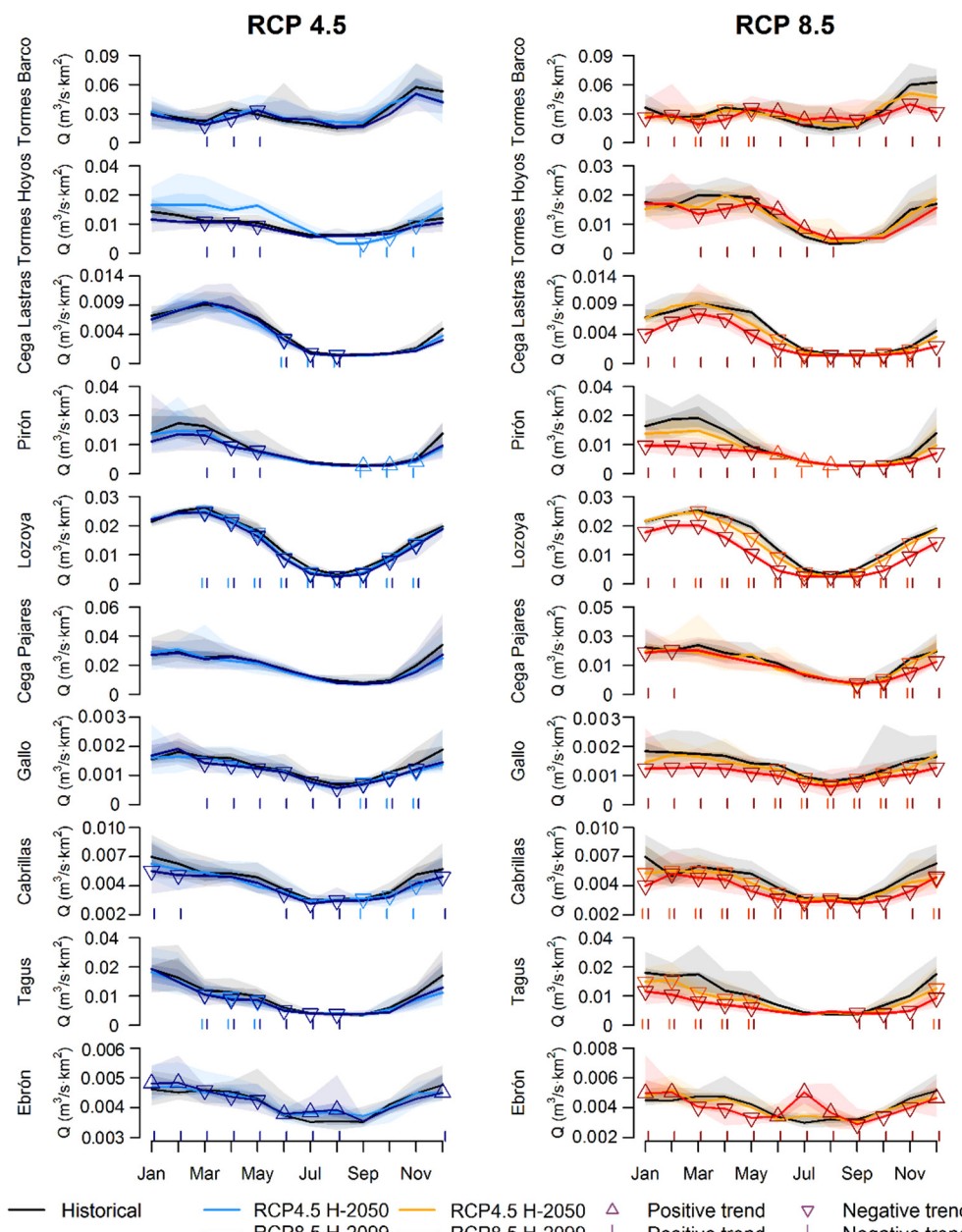

**Figure 5. Predicted monthly mean specific flow in H-2050 and H-2099 for the RCP4.5 and RCP8.5 scenarios. Shaded areas indicate decadal fluctuations. Triangles show significant negative or positive trends (Sen's slope p≤0.05); the sign of each trend is indicated by the directions in which the triangles point.**

**3.2.3 Geographical pattern**

The cluster analysis of gauging stations based on seasonal variations in the flow regime revealed the importance of careful examinations at the local level, since hydrological behaviour is a consequence of both macroclimatic and mesoclimatic

conditions. A geographical pattern is recognizable when the actual flow regime (2006-2015) is seasonally clustered (Fig. 6). Analysing the deviations in this geographical pattern by scenarios and horizons, the different gauging stations can grouped according to the seasonal behaviour of the flow changes (Fig. 7a). For the RCP4.5 scenario in H-2050 (agglomerative coefficients, a.c.= 0.73), the stations that differed most strongly from the remainder in terms of their deviations in the flow regime are those located at Cega-Lastras (winter), Pirón (autumn) and Ebrón (summer). For RCP4.5 in H-2099 (a.c.= 0.56), they are Cega-Pajares (spring), Tormes-Hoyos (summer) and Ebrón (autumn). For RCP8.5 in H-2050 (a.c.= 0.61), they are Pirón (spring, summer and autumn), Lozoya and Ebrón (both in winter). For RCP8.5 in H-2099 (a.c.= 0.72), they are Cega-Pajares (spring) and Ebrón (summer, autumn and winter).

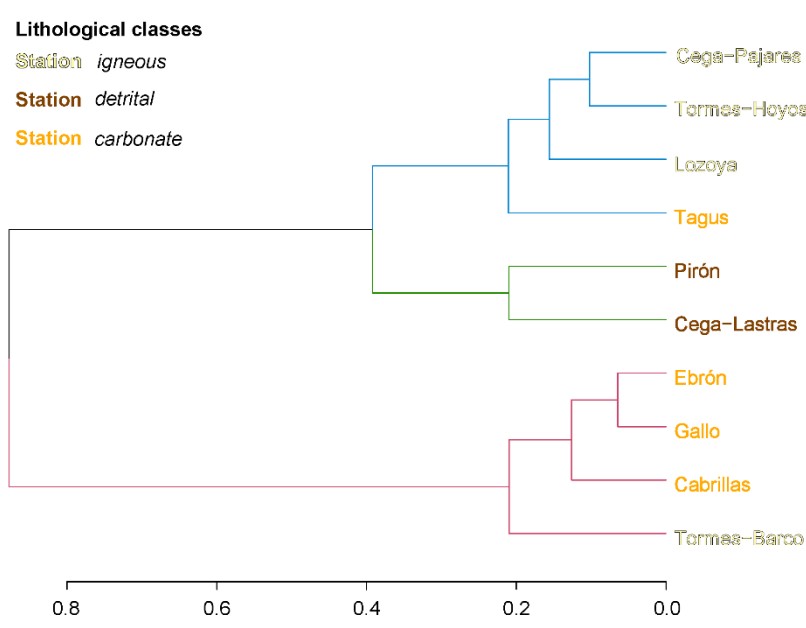

**Figure 6. Gauging stations clustered by the current normalized seasonal streamflow regime (agglomerative coefficient, a.c.= 0.81). Stations are grouped by lithological classes.**

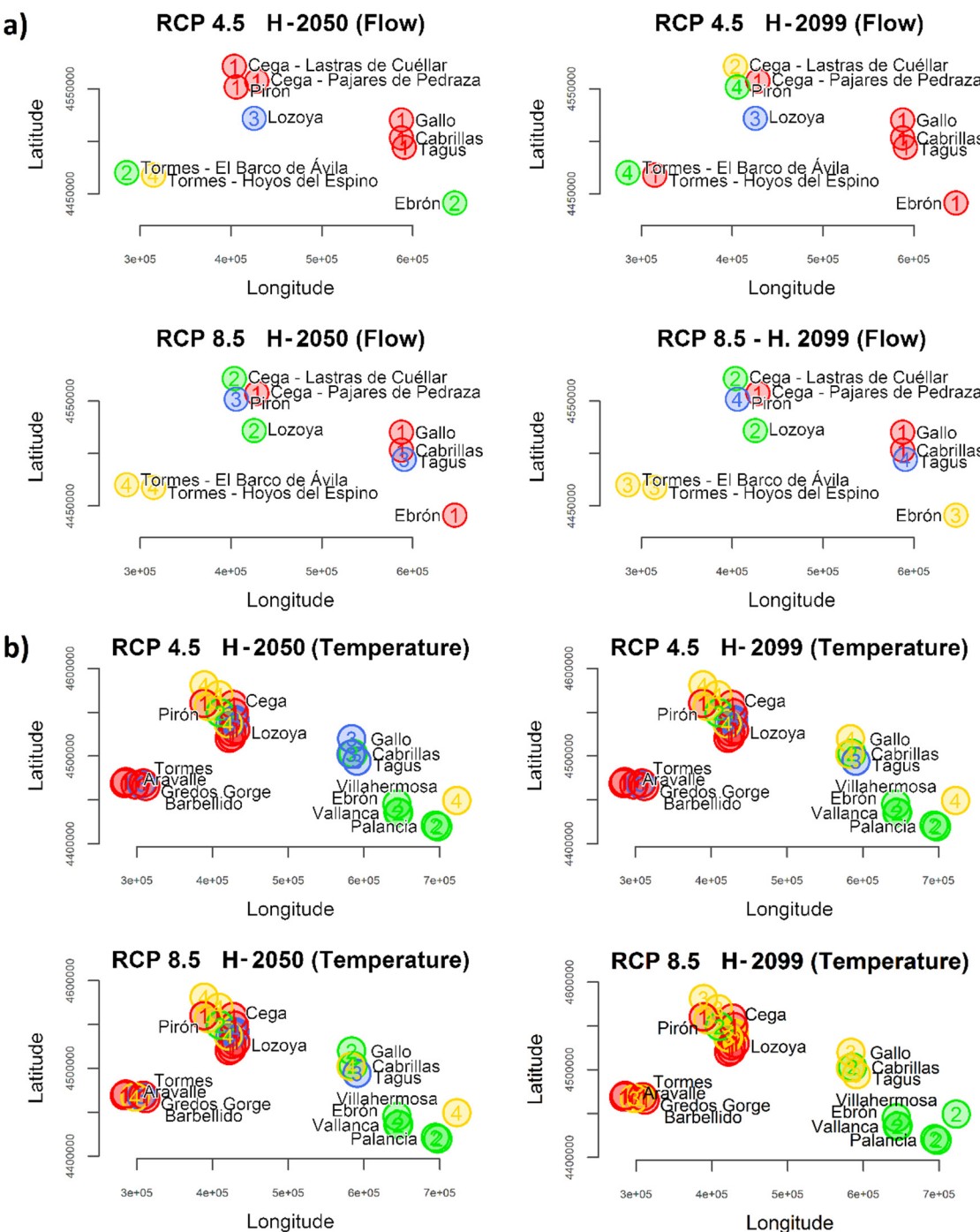

**Figure 7. Study sites clustered by the predicted change ratios of the seasonal mean streamflow (gauging stations) and by the predicted increase in the monthly mean stream temperature (°C) at the water temperature recording sites in H-2050 and H-2099 for the RCP4.5 and RCP8.5 scenarios. Axes indicate geographic positions (UTM coordinates). The colours and numbers indicate the clusters.**

### 3.3 Stream temperature

#### 3.3.1 Model parameter behaviour and general trends

The inclusion of the streamflow component improves model performance at 12 out of the 28 study sites (Table 5). In the remaining 16 cases, either no convergence of values was observed in the regression process, or the obtained values did not

improve the results, as the streamflow component (Term 2 of the equation) is virtually zero at the other sites. The five-parameter model was used in these remaining 16 cases. The calculated parameters and the performance indicators (RSE and NSE) of the models are shown in Table 6, and daily mean stream temperatures estimated by the climate change models are given in the Supplementary Material 5 (Dataset S3). Performance was high in all the cases excepting in Pirón 5 where NSE was low.

For the entire array of sites ($n$= 31), the Pearson correlation between thermal amplitude ($\alpha$-$\mu$) and $\lambda$ is significant (r-Pearson= -0.832; p<0.0001). As the thermal amplitude increases, $\lambda$ becomes more negative (indicating less resistance). As the thermal amplitude decreases, $\lambda$ approaches zero (indicating more resistance).

By the end of the XXI century, the predicted average increase in the mean annual stream temperature among the sites is 1.1°C for the RCP4.5 scenario (range 0.3-1.6°C) and 2.7°C for RCP8.5 (range 0.8-4.0°C). The average increases in maximum annual

mean temperature are predicted to be 0.8°C for RCP4.5 (range 0.1-1.5°C) and 1.6°C for RCP8.5 (range 0.2-3.0°C), and the average increases in minimum annual mean temperature are predicted to be 1.0°C (range 0.4-1.8°C) and 2.7°C (range 1.1-4.5°C), respectively. The most important increases are predicted to occur in winter, with summer experiencing smaller increases.

#### 3.3.2 Stream temperature and geological nature

The values of the model parameters showed different behaviours depending on the lithology found in each basin, which thus influences the thermal response to climate change. The thermal amplitude is greater at sites underlain by igneous bedrock ($\overline{\alpha - \mu}$= 20.38°C) than at sites underlain by carbonate bedrock ($\overline{\alpha - \mu}$= 13.07°C). $\beta$ values are greater at sites underlain by igneous bedrock ($\bar{\beta}$= 12.71°C) than at sites underlain by carbonate bedrock ($\bar{\beta}$= 7.80°C), and $\lambda$ is significantly greater ($\bar{\lambda}$= -0.140) at sites underlain by carbonate bedrock than at sites underlain by igneous bedrock ($\bar{\lambda}$= -0.292) and at sites underlain by

Quaternary detrital material ($\bar{\lambda}$= -0.305) (Fig. 8). All of these differences are significant (p-values < 0.001, as determined using t-tests with the Bonferroni correction).

Among the 8-parameter models ($n$=12), significant differences are also found among the lithological classes for $\omega$ and $\tau$. For both $\omega$ and $\tau$, the values were higher at sites underlain by carbonate bedrock ($\bar{\omega}$= 0.96°C; $\bar{\tau}$= 3.640 m$^3$·s$^{-1}$) than at sites underlain by igneous bedrock ($\bar{\omega}$= -2.12°C; $\bar{\tau}$= 0.345 m$^3$·s$^{-1}$). The differences in the $\delta$ values among the carbonate ($\bar{\delta}$= 67.06 m$^{-3}$·s) and

igneous sites ($\bar{\delta}$= 67.06 m$^{-3}$·s) were only marginally significant (p<0.1).

Under the RCP4.5 scenario, $\Delta T_{min}$ displays significantly different behaviour at sites underlain by Quaternary detrital material than at sites underlain by carbonate and igneous rocks. Under the RCP8.5 scenario, this difference is solely found between the

sites underlain by Quaternary detrital material and those underlain by carbonate rocks. $\Delta T_{\text{mean}}$ exhibits significant differences between the sites underlain by all three lithologies in both scenarios. All of these results are common to H-2050 and H-2099. In terms of $\Delta T_{\text{max}}$, in H-2050, significant differences are found between the sites underlain by carbonate and igneous rocks for both scenarios. These differences are also significant in H-2099 for RCP4.5 and RCP8.5 and between the sites underlain by

carbonate rocks and Quaternary detrital material under the RCP8.5 scenario (Fig. 9).

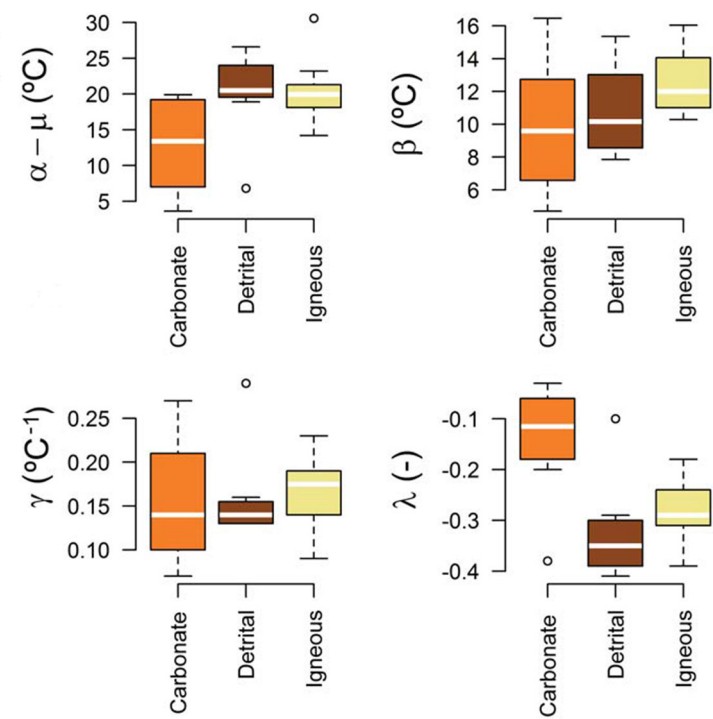

**Figure 8. Distributions of the stream temperature model parameter values (α-μ, β, γ and λ) in relation to lithology. Differences were assessed using Student's t test with the Bonferroni correction (p<0.05).**

The results of the cluster analysis of the monthly mean stream temperatures revealed a highly homogeneous aggregation of

sites for the different combinations of horizons and scenarios, given that the thermal responses of the rivers and streams are tightly linked with lithology (Fig. 7b). The carbonate sites from the Cabrillas stream (in the east) and Pirón 3 (which is strongly influenced by a calcareous spring) form a group of sites that shows low thermal amplitude and in which λ is close to zero. At the other extreme, a group that is made up mainly of sites underlain by igneous material (in the Lozoya and Tormes basins, in addition to several sites found in the detrital basin of Cega-Pirón) shows higher thermal amplitude and lower values of λ than

the former group. The remaining sites have intermediate values of thermal amplitude and resistance.

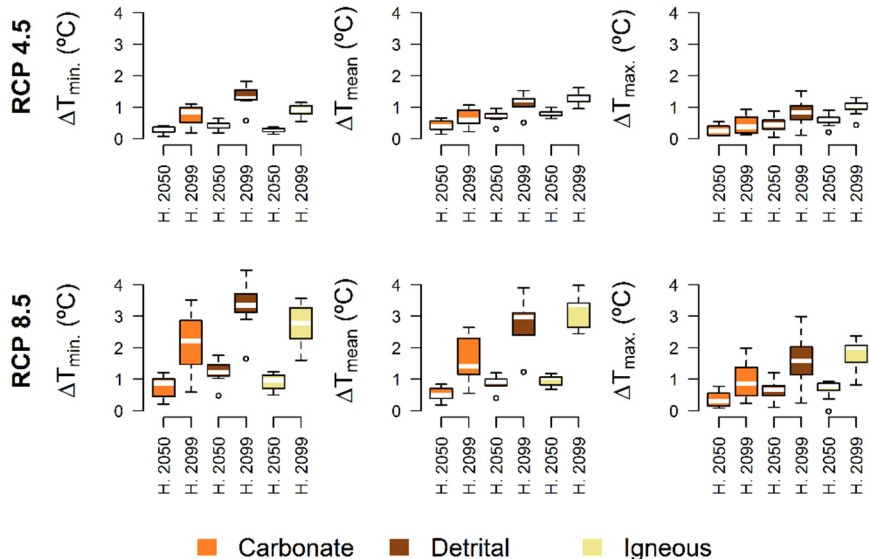

**Figure 9. Distributions of ΔTmin, ΔTmean and ΔTmax in relation to lithology for the climate change scenarios RCP4.5 and RCP8.5 in H-2050 and H-2050. The reference period corresponds to the simulated period 2010-2019.**

### 3.3.3 Effect of streamflow reductions on stream temperature

The predicted flow reductions lead to notable increases in water temperature. The effect of streamflow variation on stream temperature is analysed at the following sites: Tormes 2, Tormes 3, Pirón 1, Cega 1, Lozoya 1 to 4, Cabrillas, Ebrón 1 and Vallanca 1 and 2. These are the sites at which the 8-parameter model improves upon the 5-parameter model. In all cases, differences in stream temperature between the 5- and 8-parameter models are found, and summer flow reductions lead to increases in stream temperature, increasing DAT, TAT≥7 and MCDAT. Among these sites, the threshold is only surpassed at Lozoya and Tormes, increasing the thermal habitat loss. At Cega 1, Cabrillas and Ebrón, $\alpha$ is below the thermal threshold, and at Pirón 1, the stream temperature increase is not sufficient to exceed the threshold.

For all of the sites at which the influence of streamflow on stream temperature was revealed, the 8-parameter model estimates higher values of maximum annual DMST than the 5-parameter model. The maximum annual DMST calculated by the 8-parameter model is 3.6°C higher than that calculated by the 5-parameter model at the Tormes 2 site. This difference is not so large at the other sites, and the minimum disagreement between the models (0.01°C) is noted at the Ebrón and Cabrillas sites. In general, the maximum differences between the two models are noted in igneous catchments, whereas carbonate sites yield the lowest differences.

### 3.3.4 Effect of climate change on the thermal habitat of brown trout

The length of the thermal habitat of trout will undergo important reductions due to the rises in water temperatures and the increase in the extent of the warm period. In the predictions for H-2050, the 18.7°C threshold (TAT≥7) will be violated at eight

sites under the RCP4.5 scenario and six sites under the RCP8.5 scenario. In H-2099, the threshold will be violated at eight sites under the RCP4.5 scenario and 13 sites under the RCP8.5 scenario.

By the end of the century (H-2099), the most notable increases in TAT≥7 (Fig. 10) will be produced at Cega 6, Pirón 5 and Lozoya 3 under the RCP4.5 scenario and at Tormes 1, Cega 4, Cega 6, Lozoya 2, Gallo and Tagus-Poveda under the RCP8.5 scenario. The most significant increases in MCDAT (Fig. 10) will occur at low altitude sites underlain by igneous rocks and detrital material. In general, the highest temperatures (maximum values of 24.5°C, Table 7) are predicted to occur in the downstream reaches of the igneous and detrital river basins. In the carbonate basins, only two sites (Tagus-Poveda and Gallo) will exceed the thermal threshold. At mid-century (H-2050), the main changes under the RCP8.5 scenario are similar to those predicted for RCP4.5 at the end of the century (H-2099). RCP4.5 predicts a slower warming from mid-century onwards, whereas RCP8.5 predicts an acceleration of the warming during that period.

Continuous modelling of water temperature by means of the interpolation of model parameters along the Cega, Pirón and Lozoya rivers and the application of the model to DEM data predicts relevant losses of thermal habitat, which will affect up to 56 %, 11 % and 66 % of the lengths of these streams, respectively. In the Cega and Pirón rivers, the habitat loss is expressed relative to the proportion of total stream length where trout currently dwell (98 and 77 km in the Cega and Pirón streams, respectively). In the Lozoya River, the loss is predicted to occur in the reach (20 km) immediately upstream of a large reservoir (the Pinilla reservoir), which produces a total disconnection of the stream. The losses in maximum usable habitat will shift the current downstream limit of the trout distribution from 820 m a.s.l. up to 831 m a.s.l in the Pirón River, from 730 m a.s.l. up to 830 m a.s.l. in the Cega River, and from 1090 m a.s.l. up to 1276 m a.s.l. in the Lozoya River. In the particular case of the Cega River, a window of usable thermal habitat is also predicted to occur upstream from this altitudinal range (from 913 m a.s.l. up to 1050 m a.s.l.).

# 4 Discussion

## 4.1 Climate change

Our downscaled results predict greater air temperature increments than the original IPCC (2013) results. These higher temperatures may lead to increased ecological impacts (Magnuson and Destasio, 1997; Angilletta, 2009) caused by the combination of rising water temperatures and decreasing stream flows. The results from the AR5 of the IPCC and its annex, the Atlas of Global and Regional Climate Projections (IPCC, 2013) suggest that droughts are unlikely to increase in the near future for the Mediterranean area. However, air temperatures are expected to rise, subsequently increasing evapotranspiration. As a consequence, the available water in rivers and streams will be reduced. Regional studies have used coarser resolutions than ours, which may be appropriate for their goals (*e.g.*, Thuiller *et al.*, 2006). However, they may be insufficient when more local predictions are needed, as does our study, which treats geographically confined, stream-dwelling trout populations. Therefore, fine downscaling techniques like those applied in this study must be used when high-resolution, detailed predictions are needed.

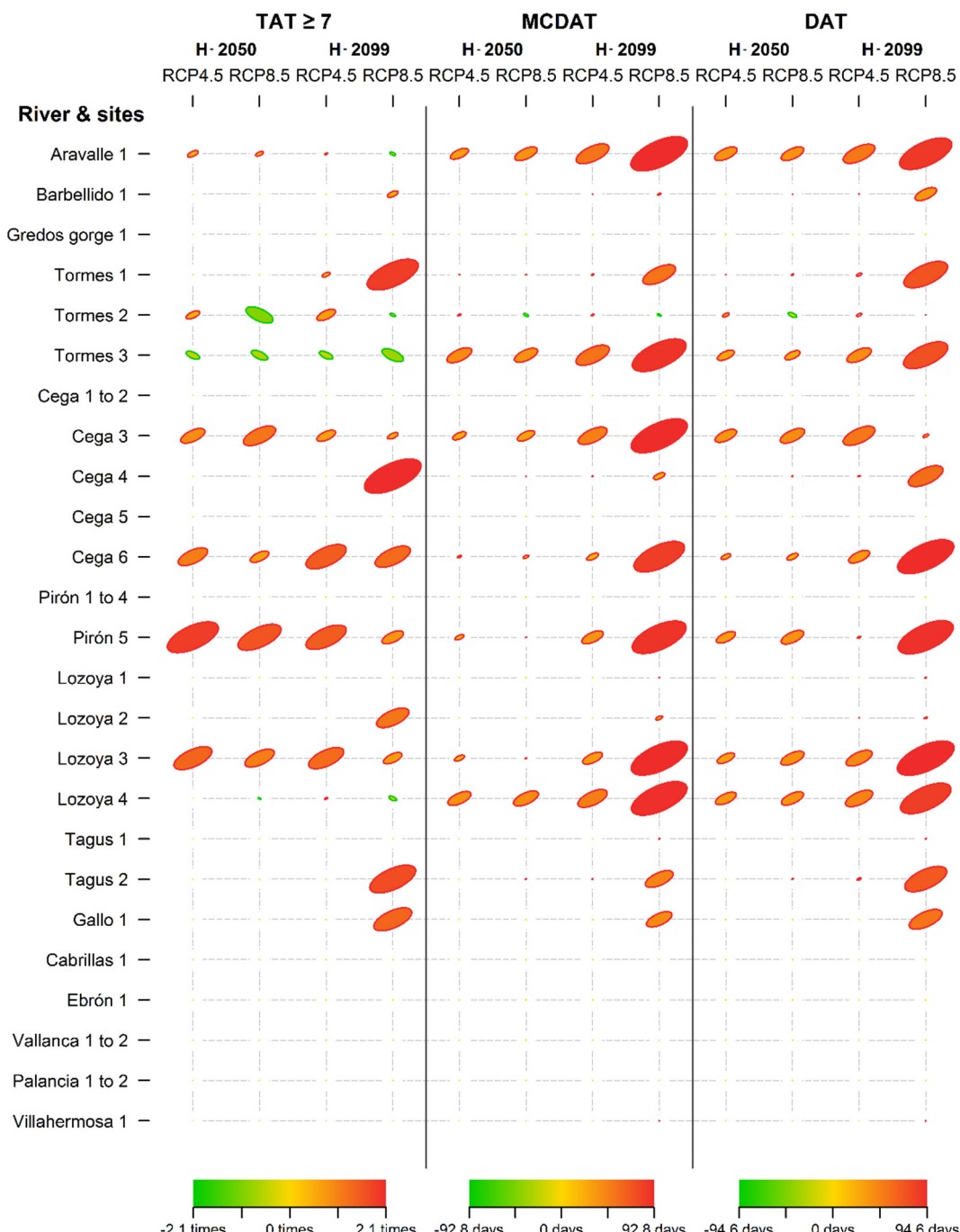

**Figure 10. Increases in TAT≥7 (time above the threshold during seven or more consecutive days), MCDAT (maximum consecutive days above the threshold) and DAT (days above the threshold per annum) from the present to H-2050 and H-2099 for RCP4.5 and RCP8.5.**

**4.2 Streamflow**

This study predicts significant but diverse streamflow reductions during the present century. At the regional level, a reduction in water resources is expected in the Mediterranean area (IPCC, 2013). Milly *et al.* (2005) predicted a 10-30 % decrease in runoff in Southern Europe in 2050. In another global-scale study, van Vliet *et al.* (2013) predicted a decrease in the mean flows of greater than 25 % in the Iberian Peninsula area by the end of the century (2071-2100), using averages for both the SRES A2 and B1 scenarios (Nakicenovic *et al.*, 2000). Our results predict mean flows that are similar to that value (-23 %, range: 0-49 %), although the emissions scenarios in this study are more severe (that is, they involve greater increases in atmospheric $CO_2$) than those used in the aforementioned studies.

More specifically, the predictions for the RCP4.5 scenario show flow reductions that range from negligibly small to significant (up to 17 %). Under the RCP8.5 scenario, significant reductions become more widespread, ranging up to 49 % of the annual streamflow losses. Our results also predict a relevant increase in the number of days with zero flow for some stations in the detrital area under this scenario (RCP8.5). The predicted streamflow changes are compatible with those obtained in previous studies, although these studies were performed at larger scales (as cited: Milly *et al.*, 2005; van Vliet *et al.*, 2013). The apparent differences between the streamflow reductions estimated in this study and those obtained by Milly *et al.* (2005) and van Vliet *et al.* (2013) (who report lower flow reductions than those given in the present study) might be caused by the regional focus of their predictions (the entire Iberian Peninsula), whereas ours are focused on mountain reaches.

In terms of methods, process-based hydrological models are often preferred for climate change studies (Van Vliet *et al.*, 2012). However, they can be overly complicated and require excessive data inputs, which may also lead to over-fitting of the data (Zhuo *et al.*, 2015). Constraining further predictions to within the training domain is a rule of thumb for machine learning studies (Fielding, 1999), although extrapolation is rather common (Elith and Leathwick, 2009). Therefore, taking into account the extrapolation that occurs towards lower flows, which are overrepresented in the training dataset, we consider the magnitude of the extrapolation acceptable, and we consider the values, although they are not exempt from uncertainty, to be reliable.

**4.3 Stream temperature**

The model we present in this study showed good performance. Bustillo *et al.* (2013) recommended the assessment of the impacts of climate change on river temperatures using regression-based methods like ours that rely on logistic approximations of *equilibrium temperatures* (Edinger *et al.*, 1968), which are at least as robust as the most refined classical heat balance models.

However, we also sought to identify relationships between thermal regime and other environmental variables besides air temperature and streamflow, such as geology. Bogan *et al.* (2003) showed that water temperatures were uniquely controlled by climate in only 26 % of 596 studied stream reaches. Groundwater, wastewater and reservoir releases influenced water temperatures in the remaining 74 % of the cases. Loinaz *et al.* (2013) quantified the influence of groundwater discharge on temperature variations in the Silver Creek Basin (Idaho, USA), and they concluded that a 10 % reduction in groundwater flow

can cause increases of over 0.3°C and 1.5°C in the average and maximum stream temperatures, respectively. Our studied reaches were not influenced by wastewater or reservoir releases (with the exception of releases from the Torrecaballeros Dam on the Pirón River). Kurylyk *et al.* (2015) showed that the temperature of shallow groundwater influences the thermal regimes of groundwater-dominated streams and rivers. Since groundwater is strongly influenced by geology, we can expect it to be a good indicator of the thermal response, as shown here. The models used accurately described the thermal performance of the study sites, and we found significant relationships among the model parameters, the underlying lithologies and the hydrologic responses. Thermal amplitude ($\alpha$-$\mu$) and temperature at the maximum change rate ($\beta$) were lower, and the resistance parameter ($\lambda$) was closer to zero, in river basins that were highly influenced by aquifers (mainly carbonate) compared to the others, particularly compared with river basins underlain by carbonate rocks. Since DMST is a variable that is relevant for detecting departures from thermal niche, we can conclude that it is worthwhile to use the more complex 8-parameter model to predict the effects of global warming, especially in igneous catchments.

A wide range of models is described in the literature, and each such model has its strengths and weaknesses. Arismendi *et al.* (2014) hold that regression models based on air temperature can be inadequate for projecting future stream temperatures because they are only surrogates for air temperature, whereas Piccolroaz *et al.* (2016) argued that the adequacy depends on the hydrological regime, type of model and the time scale analysis. Their main objections to regressive methods arose when modelling reaches of regulated rivers, but this is not our case. Besides, our model improves the models that were tested in both studies (Arismendi *et al.*,2014, Piccolroaz *et al.*, 2016). Performance indicators of our models produce good results showing that models are sufficiently competent. We show that our model implicitly integrates the effect of other factors, such as geology and flow regime by means of its parameters. Fine mechanistic solution to the modelling issue could need prohibitive methods (Kurylyk *et al.*, 2015) losing the advantages that make attractive the model (input data easy to get). Therefore a compromise between improved precision and increased cost must be met.

The behaviour and dynamics of the parameters offer a promising research field. Their analysis may help to introduce new parametrization criteria to avoid the risk of ignoring the effect of climate warming on groundwater (subsurface water and deep water), for instance. The thermal sensitivity of shallow groundwater differs between short-term (*e.g.*, seasonal) and long-term (*e.g.*, multi-decadal) time horizons, and the relationship between air and water temperatures does not necessarily reflect this difference. This variability should be taken into account in order to avoid underestimating the effects of climate warming (Kurylyk *et al.*, 2015).

Regression models are substantially site specific compared to deterministic approaches (Arismendi *et al.*, 2014). However, the parameters of these regression approaches are still physically meaningful, and these models require fewer variables that can limit the applicability of more complex models in areas where data are scarce. Consequently, the value of this type of model is its applicability to a large number of sites where the only available data describe air temperatures (and precipitation and streamflow to a lesser extent). On the other hand, our results show that predictions can improve when streamflow is included in the water temperature model, although some streams show little or no sensitivity to the introduction of streamflow into the model. However, the lack of sensitivity is not necessarily be due to the absence of the influence of flow on the water

temperature but rather to its minor relevance compared to other sources of noise. Thus, when flow data are available, it may be recommended to use the more complex 8-parameter model to predict the effects of climate warming. This conclusion is especially applicable to lithologically sensitive basins, such as those underlain by igneous rocks.

The predicted increase in water temperature will be substantial at most of the study sites. The annual mean rates of change will increase with time. Stewart *et al.* (2015) predict an increase of 1-2°C by mid-century in 80 % of the stream lengths in Wisconsin and by 1-3°C by the latter part of the century in 99 % of the stream lengths, which corresponds to a significant loss in suitable areas for cold-water fish. The results of Stewart *et al.* (2015) do not differ from ours, except that we expect greater increases by the end of the studied period (up to 4°C). Our results are also compatible with those of van Vliet *et al.* (2013), who predicted a water temperature increase >2°C in the Iberian Peninsula area by the end of the century; however, our results are more specific and precise. In this sense, Muñoz-Mas *et al.* (2016) also obtained similar results for H-2050 in a river reach in central Spain by mid-century (*i.e.*, daily mean flow reductions between 20-29 % and daily mean stream temperature increases up to 0.8°C). However, we predict that the minima are more sensitive to climate warming than the maxima.

### 4.4 Effects of climate change on brown trout populations

Brown trout are sensitive to changes in discharge patterns because high intensity floods during the incubation and emergence periods may limit recruitment (Lobón-Cerviá and Rincón, 2004; Junker *et al.*, 2015). In the Iberian Peninsula, the trout distribution is mainly concentrated in mountain streams, where extreme discharges during winter are expected to increase (Rojas *et al.*, 2012). These extreme discharges will likely affect trout recruitment negatively. Thus, the predicted changes in the hydrological regime can subject brown trout populations to more variable conditions, which may occasionally present some populations with insuperable bottlenecks. Trout are polytypic and display an adaptable phenology and rather high intra-population variability in their life history traits that might allow them to show resilience to variations in habitat features (Gortázar *et al.*, 2007; Larios-López *et al.*, 2015), especially in the marginal ranges (Ayllón *et al.*, 2016). However, despite these strong evolutionary responses, the current combination of warming and streamflow reduction scenarios is likely to exceed the capacity of many populations to adapt to new conditions (Ayllón *et al.*, 2016). Consistent with regional predictions (Rojas *et al* 2012; Garner *et al.*, 2015), significant flow reductions are expected during summertime in most of the studied rivers and streams at the end of the century, and this may mean, in turn, the reduction in the suitable habitat (*i.e.*, the available water volume) (Muñoz-Mas *et al.*, 2016). Finally, the increase in extreme droughts, which involve absolute water depletion, in certain reaches of the streams may be critical for some trout populations.

The predicted increase in winter stream temperatures can affect the sessile phases (*i.e.*, eggs and larvae) of trout development. These phases are very sensitive to temperature changes because it affects their physiology, and because their development is temperature dependent (*e.g.*, Lobón-Cerviá and Mortensen, 2005; Lahnsteiner and Leitner, 2013). Thus, changes in the duration of incubation and yolk sac absorption can affect emergence times and, in turn, the sensitivity of these phases to hydrological regime alterations (Sánchez-Hernández and Nunn, 2016). An increase in stream temperature can also reduce hatchling survival (Elliott and Elliott, 2010). In accordance with the results presented herein, the predicted synergy of

streamflow reductions and water temperature increases will cause substantial losses of suitable fish habitat, especially for cold-water fish such as brown trout (Muñoz-Mas *et al.*, 2016).

The increases in threshold violations were important in our simulations. The duration of warm events (temperature above the threshold value) increased by up to three months at the end of the century in the most pessimistic scenario (RCP8.5). A continuous analysis of the whole-river response should be conducted to allow spatially explicit predictions and to identify reaches where thermal refugia are likely to occur. However, our results suggest that trout will not survive in these reaches because the persistence of thermal refugia is improbable or because their extents will be insufficient. In the Cega, Pirón and Lozoya rivers, important losses of thermal habitat will occur that could jeopardize the viability of the trout population. Behavioural thermoregulatory tactics are common in fish (Reynolds and Casterlin, 1979; Goyer *et al.*, 2014); for instance, some species perform short excursions (<60 min in experiments with brook char, *S. fontinalis*) that could be a common thermoregulatory behaviour adopted by cold freshwater fish species to sustain their body temperature below a critical temperature threshold, enabling them to exploit resources in an unfavourable thermal environment (Pépino *et al.*, 2015). Brown trout can use pool bottoms during daylight hours to avoid the warmer and less oxygenated surface waters in thermal refugia (Elliott, 2000). Nevertheless, if the warm events became too long, the thermal refugia could become completely insufficient, thus compromising fish survival (Brewitt and Danner, 2014; Daigle*, et al.*, 2014).

## 4.5 The brown trout distribution

According to our results, streamflow reductions are able to synergistically contribute to the loss of thermal habitat by increasing daily mean stream temperatures. This effect is especially relevant in summer in the Mediterranean area, when the warmest temperatures and minimum flows usually occur. The existence of thermal refugia represents a possible means of fish survival, and the probability for a water body to become a thermal refugium is highly geologically dependent. In our simulations, the sites that are most dependent on deep aquifers (*i.e.*, basins underlain by Mesozoic carbonate rocks) display improved resistance to warming. The habitat retraction at the rear edge of the actual distribution of brown trout is deduced to be geologically mediated.

The mountains of central and southeastern Spain contain the rear edge of the distribution of native brown trout (Kottelat and Freyhof, 2007). Fragmentation and disconnection of populations by newly formed thermal barriers may aggravate the already significant losses of thermal habitat by reducing the viability of populations and increasing the extinction risk. Thus, the rear edge of the trout population in the Iberian Peninsula might shift to the northern mountains to varying extents depending on the presence of relevant mesological features, such as geology. The calcareous mountains of northern Spain could be a refuge for trout because they combine favourable geology and a relatively more humid climate. Caused by this differential response, the western portion of the Iberian range (which is plutonic and less buffered) will eventually experience more frequent local temperature-driven extinction events, thus producing a greater shift northward, than in the Eastern Iberian end of this range, which is calcareous and highly buffered and will remain more resilient to these local extinction events. However, the predicted streamflow reductions may act synergistically, reducing the physical space, and this may jeopardize the less thermally exposed

populations. In the Iberian Peninsula, stream temperatures will increase less in the central and northern mountains than in the central plateau, and the increases will be smaller in karstic than in granitic (igneous) mountains. At the same time, the side of the peninsula that faces the Mediterranean is expected to be more sensitive to warming and streamflow reductions than the side of the peninsula that faces the Atlantic. Thus, brown trout populations in the karstic mountains of northern Spain (the Cantabrian Mountains and the calcareous parts of the Pyrenees) are better able to resist the climate warming than the populations farther east in the granitic portion of the Pyrenees (Santiago 2017). Similar patterns may occur in other parts of Southern Europe. Most likely, the less pronounced thermal responses of rivers and streams in the karstic areas will allow for greater persistence of the brown trout population, although changes in streamflow regimes will likely also occur there.

In a study of the major basins of Europe, Lassalle and Rochard (2009) predicted that the brown trout would "lose all its suitable basins in the southern part of its distribution area ([the] Black Sea, the Mediterranean, the Iberian Peninsula and the South of France), but [would] likely to continue being abundant in [the] northern basins". Almodóvar *et al.* (2011) estimated that the brown trout will be eradicated over almost the entire stream length of the studied basins in North Spain, and Filipe *et al.* (2013) estimated an expected loss of 57 % of the studied reaches in the Ebro basin in north-eastern Spain. Our study shows important, yet not so dramatic, reductions in the thermal habitat of Iberian brown trout populations in mountainous areas. The number of general climate models used, the reliability of the downscaling procedure, the resolution of the stream temperature and streamflow models, and the method used to study the threshold imply a substantial improvement in detail (Santiago *et al.*, 2016) over previous work. It is reasonable to infer that many mountain streams appear poised to become refugia for cold-water biodiversity during this century (Isaak *et al.*, 2016).

## 5 Conclusions

The main findings of this study are as follows. (i) Our downscaled results predict greater air temperature increments than the IPCC's averages, from which our estimations were made; (ii) significant but diverse streamflow reductions are predicted to occur during the present century; (iii) the models presented in this study have been shown to be useful for improving simulations; (iv) the predicted increases in water temperature will be influenced to varying degrees by the flow and geological features of rivers and streams; (v) the thermal habitat of brown trout, a cold-water species, will decrease as a consequence of the synergistic effects of flow reduction and water warming; and (vi) the peaks in water temperature and the complete depletion of the river channels will produce local extinctions, although the ultimate magnitude of the effect will be governed by the geological nature of the basins.

Our findings might be useful in planning the prevention and mitigation of the negative effects of climate change on freshwater fish species at the rear edge of their distributions. A differentiation of areas based on their risk level and viability is necessary to set standardized conservation goals. Our results show that trout conservation requires knowledge of both temperature and streamflow dynamics at fine spatial and temporal scales. Managers need easy-to-use tools to simulate the expected impacts

and the management options to address them, and the methods and results we provide could provide key information in developing these tools and management options.

**Acknowledgments:** We are grateful to the Consejería de Medio Ambiente y O.T. of the Government of Castilla y León, especially to Mariano Anchuelo, Fabián Mateo and the forest ranger team of Navafría. Also, we are in debt with the Sierra of Guadarrama National Park staff, and especially with Juan Bielva and Ángel Rubio for the temperature data of Lozoya stream. Juan Diego Alcaraz is the author of the temperature data of the Ebrón basin. Valérie Ouellet and two anonymous referees provided valuable comments that substantially improved the original manuscript. The World Climate Research Programme's Working Group on Coupled Modelling is responsible for the 5[th] Coupled Model Intercomparison Project, and we thank the climate modelling groups for producing and making available their model output. Climate change study was partially funded by the Ministerio de Agricultura, Alimentación y Medio Ambiente of Spain through the Fundación para la Investigación del Clima (http://www.ficlima.org/), by the European Union, DG for Environment (DURERO Project. C1.3913442), by the Ministerio de Economía y Competitividad of Spain (IMPADAPT project CGL2013-48424-C2-1-R) and FEDER funds.

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

**Appendix 1: Table of symbols and acronyms.**

Alphabetically ordered.

| symbol/acronym | meaning |
| --- | --- |
| a.c. | agglomerative coefficient |
| AEMET | Spanish Meterorological Agency |
| AIC | Akaike information criterion |
| $\alpha$ | maximum stream temperature |
| AR5 | Fifth Assessment Report of the IPCC |
| $\beta$ | air temperature at which the rate of change of the stream temperature with respect to the air temperature is a maximum |
| BIC | Bayesian information criterion |
| CMIP5 | Fifth Coupled Model Intercomparison Project |
| DAT | total Days Above the Threshold per year |
| $\delta$ | maximum rate at $\tau$. |
| $\Delta Ta$ | 1-day variation of the daily mean air temperature |
| $\Delta T_{max}$ | increments of the annual averages of the daily maximum stream temperature |
| $\Delta T_{mean}$ | increments of the annual averages of the daily mean stream temperature |
| $\Delta T_{min}$ | increments of the annual averages of the daily minimum stream temperature |
| DEM | digital elevation model |
| DMAT | daily mean air temperature |
| DMST | daily mean steam temperature |
| ECDF | Empirical Cumulative Distribution Functions |
| $\gamma$ | rate of change at $\beta$ |
| H-2050 | year horizon 2050 |
| H-2099 | year horizon 2099 |
| IGN | National Geographic Institute of the Spanish Government |
| IPCC | Intergovernmental Panel on Climate Change |
| $\lambda$ | represents the resistance of DMST to change with respect to the 1-day variation in DMAT ($\Delta Ta$) |
| M5 | Machine learning technique |
| MCDAT | Maximum Consecutive Days Above the Threshold per year |
| $\mu$ | minimum stream temperature |
| NSE | Nash-Sutcliffe Efficiency |
| $\omega$ | maximum observable variation in stream temperature due to the flow difference |
| PACF | partial autocorrelation function |
| Q10 | tenth percentile of flow |
| $Q_{max}$ | maximum flow |
| $Q_{mean}$ | mean flow |
| $Q_{min}$ | minimum flow |

| | |
|---|---|
| RCP4.5 | Representative Concentration Pathway 4.5 |
| RCP8.5 | Representative Concentration Pathway 8.5 |
| RSE | Residual Standard Error |
| Ta | air temperature |
| TAT≥7 | frequency of events of seven or more consecutive days above the threshold per year |
| τ | low value at which the rate of change of the stream temperature with respect to the flow is a maximum |
| Ts | stream temperature |

## Appendix 2: Uncertainty.

In science, and particularly in hydrological studies, the uncertainty is a matter which requires special attention and it has led us to include a synthesis of our approach to this problem in this appendix. The uncertainty analysis is a necessary step in assessing the risk level in the applicability of a model (Pappenberger and Beven, 2006). Our aim was to study the viability of the brown trout populations, and we modelled the flow and the stream temperature for this purpose. From a conceptual point of view our approaches were consistent with it.

On data inputs to build the models, uncertainties and inconsistencies are a habitual issue (Juston *et al*., 2013). The Meteorological and Hydrological Services subject data to their own quality controls but systematic error cannot always be completely controlled (Beven and Westerberg, 2011; McMillan *et al*., 2012). For this reason, in addition, we tested the input data seeking inconsistencies.

The modelling of the river reaches as one-dimensional elements implies a simplification of the fluvial ecosystem that is generally accepted at this scale (e.g., Viganò *et al*., 2015; Ahmed and Tsanis, 2016), especially for ecological purposes (e.g., Caiola *et al*. 2014). Nevertheless, the size of the rivers under study made little or nothing relevant the variations in width and depth (it was verified in the field).

Regarding the parameterization of the models, cross-validation was used to evaluate the uncertainty in these process, and indicators such as the NSE (for hydrological and thermal models), the deviance and the RSE (for thermal models) were calculated. In the case of the thermal model, the functions of distribution of the parameters of the model were built by non-parametric bootstrap, and the mean values were chosen as the most proficient estimators. As results show, parameters tell us about the functional behaviour of catchments (particularly on the effects of the catchments geology on the streams temperature) and this might improve predictions in ungauged basins by better controlling uncertainty (Juston *et al*., 2013).

Once the models were constructed, It was verified that the overlaps of the ranges of the model input variables and the ranges of the outputs were significant ($p < 0.05$). The non-overlapping zones affected, on the one hand, infrequent events (great floods) and the extreme temperature zone (zone of extrapolation), being the last one the scope in which we expected to work. However, the weakness of the hydrological model in the flood zone should be considered for other applications and developments of the model. The hydrological model given us sufficient and relevant information since its possible weak points (extrapolation in

the floods assessment) did not affect our goal: we focused on central trends and minimum values, and they were solidly represented in the samples and in the simulations.

As said, the inherent uncertainty of the climate predictions according to the scenarios RCP4.5 and RCP8.5 was attenuated by means of the ensemble technique, showing the dispersion of the results by mean of the percentiles in Fig. S1 to S24 (Supplementary Material). Beven (2011) exposed his legitimate concerns on the credibility of climate models which fail when are compared with the control period and, consequently, we used ERA-40 reanalysis to control this source of bias with excellent results.

There was a time-dependence between the errors of the model and the scope of the prediction, but these errors were only important in the zone of high temperature and low flow, as expected by the physical nature of the climatic variables. Moreover, this is the variables behaviour that was our intention to evaluate.

Ahmed, S. and Tsanis, I.: Hydrologic and Hydraulic Impact of Climate Change on Lake Ontario Tributary. Am. J. Water Resour., 4(1), 1-15. doi: 10.12691/ajwr-4-1-1, 2016.

Beven, K.: I believe in climate change but how precautionary do we need to be in planning for the future? Hydrol. Process., 25(9), 1517-1520. doi: 10.1002/hyp.7939, 2011.

Beven, K. and Westerberg, I.: On red herrings and real herrings: disinformation and information in hydrological inference. Hydrol. Process., 25(10), 1676–1680. doi: 10.1002/hyp.7963, 2011.

Caiola, N., Ibáñez, C., Verdú, J. and Munné, A.: Effects of flow regulation on the establishment of alien fish species: A community structure approach to biological validation of environmental flows, Ecol. Indic., 45, 598-604. doi: 10.1016/j.ecolind.2014.05.012. 2014.

Juston, J.M., Kauffeldt, A., Montano, B.Q., Seibert, J., Beven, K.J. and Westerberg, I.K.: Smiling in the rain: Seven reasons to be positive about uncertainty in hydrological modelling. Hydrol. Process., 27(7), 1117–1122. doi:10.1002/hyp.9625, 2013.

McMillan, H., Krueger, K. and Freer, J.: Benchmarking observational uncertainties for hydrology: Rainfall, river discharge and water quality. Hydrol. Process., 26(26), 4078-4111. doi: 10.1002/hyp.9384, 2012.

Pappenberger, F. and Beven, K.J.: Ignorance is bliss: 7 reasons not to use uncertainty analysis. Water Resour. Res., 42: W05302. doi: 10.1029/2005WR004820, 2006.

Viganò, G., Confortola, G., Fornaroli, R., Cabrini, R., Canobbio, S., Mezzanotte, V. and  Bocchiola, D.: Effects of future climate change on a river habitat in an Italian alpine catchment. J. Hydrol. Eng., 21(2), 04015063. doi: 10.1061/(ASCE)HE.1943-5584.0001293, 2015.

**Table 1. Description of the data logger (thermograph) sites, specifying given name, UTM-coordinates (Europe WGS89), altitude (m above the sea level), code of the nearest temperature meteorological station with suitable time series for this study (AEMET: Spanish Meteorological Agency), orthogonal distance between the data logger and the meteorological station, number of recorded days for stream temperature and characteristic geological nature (lithology) of the data logger site (the latter was obtained from IGME [2015]). Bold letters indicate sites associated to the gauging stations.**

| Sites | UTM-X | UTM-Y | altitude (m a.s.l.) | AEMET code | distance to AEMET station (km) | recording days | lithology |
|---|---|---|---|---|---|---|---|
| Aravalle | 283623 | 4468847 | 1010 | 2440 | 76.4 | 1257 | Igneous |
| Barbellido | 311759 | 4465519 | 1440 | 2440 | 52.2 | 881 | Igneous |
| Gredos Gorge | 306363 | 4468087 | 1280 | 2440 | 55.7 | 644 | Igneous |
| **Tormes1** | **308751** | **4469371** | **1270** | **2440** | **53.0** | **421** | **Igneous** |
| Tormes2 | 297543 | 4467191 | 1135 | 2440 | 64.0 | 537 | Igneous |
| **Tormes3** | **285481** | **4470750** | **995** | **2440** | **74.1** | **588** | **Igneous** |
| Cega1 | 427627 | 4539806 | 1600 | 2516 | 84.5 | 544 | Igneous |
| Cega2 | 429416 | 4541728 | 1384 | 2516 | 85.8 | 544 | Igneous |
| Cega3 | 428892 | 4549370 | 1043 | 2516 | 83.9 | 544 | Igneous |
| **Cega4** | **426932** | **4559076** | **943** | **2516** | **81.2** | **407** | **Quaternary detrital** |
| **Cega5** | **408504** | **4569772** | **853** | **2516** | **63.4** | **544** | **Quaternary detrital** |
| Cega6 | 389014 | 4581160 | 766 | 2516 | 47.9 | 501 | Quaternary detrital |
| Pirón1 | 422082 | 4536456 | 1475 | 2516 | 80.1 | 544 | Igneous |
| Pirón2 | 420660 | 4537094 | 1348 | 2516 | 78.6 | 483 | Igneous |
| **Pirón3** | **409935** | **4549473** | **908** | **2516** | **65.2** | **544** | **Quaternary detrital** |
| Pirón4 | 394462 | 4556823 | 826 | 2516 | 48.9 | 544 | Quaternary detrital |
| Pirón5 | 388615 | 4560166 | 815 | 2516 | 42.9 | 424 | Quaternary detrital |
| Lozoya1 | 422060 | 4520319 | 1452 | 3104 | 7.3 | 2151 | Igneous |
| Lozoya2 | 425445 | 4522314 | 1267 | 3104 | 4.6 | 1870 | Igneous |
| **Lozoya3** | **425657** | **4527327** | **1142** | **3104** | **0.7** | **1776** | **Igneous** |
| Lozoya4 | 430740 | 4530050 | 1090 | 3104 | 6.4 | 2187 | Igneous |
| **Tagus-Peralejos** | **590887** | **4494165** | **1149** | **3013** | **27.9** | **964** | **Carbonate** |
| Tagus-Poveda | 582900 | 4502160 | 1028 | 3013 | 22.8 | 669 | Carbonate |
| **Gallo** | **583771** | **4519743** | **998** | **3013** | **10.9** | **1019** | **Carbonate** |
| **Cabrillas** | **585619** | **4502986** | **1075** | **3013** | **20.8** | **1070** | **Carbonate** |
| **Ebrón** | **643551** | **4445027** | **879** | **8381B** | **9.5** | **592** | **Carbonate** |
| Vallanca1 | 644966 | 4435479 | 745 | 8381B | 1.8 | 836 | Carbonate |
| Vallanca2 | 645936 | 4435715 | 718 | 8381B | 0.8 | 836 | Carbonate |
| Palancia1 | 694348 | 4421176 | 760 | 8434A | 10.4 | 334 | Carbonate |
| Palancia2 | 697451 | 4419477 | 660 | 8434A | 7.8 | 334 | Carbonate |
| Villahermosa | 722594 | 4449436 | 592 | 8478 | 13.5 | 334 | Carbonate |

**Table 2.** Official stations used (meteorological and hydrological), variables, length of time series used and geographical position. AEMET: Spanish Meteorological Agency; CHD: Water Administration of Duero Basin; CHT: Water Administration of Tagus Basin; and CHJ: Water Administration of Júcar Basin.

| Institution | code | name | variable | used series length | | UTM-X | UTM-Y | altitude |
|---|---|---|---|---|---|---|---|---|
| AEMET | 2180 | Matabuena | pluviometry | 1955- | 2013 | 436266 | 4549752 | 1154 |
| AEMET | 2186 | Turégano | pluviometry | 1955- | 2013 | 415346 | 4556596 | 935 |
| AEMET | 2196 | Torreiglesias | pluviometry | 1970- | 2013 | 413294 | 4550606 | 1053 |
| AEMET | 2199 | Cantimpalos | pluviometry | 1955- | 2013 | 402524 | 4547811 | 906 |
| AEMET | 2440 | Aldea del Rey Niño | temperature | 1955- | 2012 | 356059 | 4493201 | 1160 |
| AEMET | 2462 | Puerto de Navacerrada | temperature and pluviometry | 1967- | 2012 | 414745 | 4516276 | 1894 |
| AEMET | 2516 | Ataquines | temperature | 1970- | 2013 | 345716 | 4560666 | 802 |
| AEMET | 2813 | Navacepeda de Tormes | pluviometry | 1965- | 2012 | 308892 | 4470347 | 1340 |
| AEMET | 2828 | El Barco de Ávila | temperature and pluviometry | 1955- | 1983 | 285643 | 4470512 | 1007 |
| AEMET | 3009E | Orihuela del Tremedal | pluviometry | 1986- | 2000 | 614383 | 4489759 | 1450 |
| AEMET | 3010 | Ródenas | pluviometry | 1968- | 2006 | 625505 | 4499963 | 1370 |
| AEMET | 3013 | Molina de Aragón | temperature and pluviometry | 1951- | 2010 | 594513 | 4521786 | 1056 |
| AEMET | 3015 | Corduente | pluviometry | 1961- | 2000 | 584125 | 4523281 | 1120 |
| AEMET | 3018E | Aragoncillo | pluviometry | 1968- | 2010 | 580519 | 4531876 | 1263 |
| AEMET | 3104 | Rascafría-El Paular | temperature and pluviometry | 1967- | 2012 | 425165 | 4526895 | 1159 |
| AEMET | 8376B | Jabaloyas | pluviometry | 1993- | 2006 | 635600 | 4456215 | 1430 |
| AEMET | 8381B | Ademuz-Agro | temperature and pluviometry | 1989- | 2010 | 646722 | 4436034 | 740 |
| AEMET | 8434A | Viver | temperature | 1971- | 2006 | 704704 | 4422256 | 562 |
| AEMET | 8478 | Arañuel | temperature | 1971- | 2006 | 714943 | 4438277 | 406 |
| CHD | 2006 | Tormes-Hoyos del Espino | flow | 1955- | 2012 | 314676 | 4467908 | 1377 |
| CHD | 2016 | Cega-Pajares de Pedraza | flow | 1955- | 2013 | 428296 | 4557678 | 938 |
| CHD | 2057 | Pirón-Villovela de Pirón | flow | 1972- | 2013 | 405596 | 4551929 | 869 |
| CHD | 2085 | Tormes-El Barco de Ávila | flow | 1955- | 2012 | 285173 | 4470362 | 992 |
| CHD | 2714 | Cega-Lastras de Cuéllar | flow | 2004- | 2013 | 403509 | 4571682 | 838 |
| CHT | 3001 | Tagus-Peralejos de las Truchas | flow | 1946- | 2010 | 590474 | 4494474 | 1143 |
| CHT | 3002 | Lozoya-Rascafría (El Paular) | flow | 1967- | 2013 | 425321 | 4522069 | 1270 |
| CHT | 3030 | Gallo-Ventosa | flow | 1946- | 2010 | 587349 | 4520522 | 1016 |
| CHT | 3268 | Cabrillas-Taravilla | flow | 1982- | 2010 | 587480 | 4503395 | 1107 |
| CHJ | 8104 | Ebrón-Los Santos | flow | 1989- | 2010 | 645963 | 4441366 | 750 |

**Table 3. Different classes of thermal thresholds for emerged trout classes found in literature. The type of experiment differentiates the experiments with controlled (laboratory) and uncontrolled (wild) temperature. Latitude of the experiments' location is showed.**

| variable | temperature (°C) | type of experiment | latitude | reference |
|---|---|---|---|---|
| maximum growth | 13.1 | laboratory | 54ºN | Elliott *et al.* 1995 |
| maximum growth | 16 | laboratory | 61ºN | Forseth and Jonsson 1994 |
| maximum growth | 16.9 | laboratory | 43ºN | Ojanguren *et al.* 2001 |
| maximum growth | 13.2 | wild | 43ºN | Lobón-Cerviá and Rincón 1998 |
| maximum growth | 13 | wild | 41ºS | Allen 1985 |
| maximum growth | 15.4-19.1 | laboratory | 59ºN | Forseth *et al.* 2009 |
| thermal optimum | 14.2 | wild | 47ºN | Hari *et al.* 2006 |
| upper growth limit | 19.5 | wild | 41ºS | Allen 1985 |
| upper thermal niche | 20 | wild | 47ºN | Hari *et al.* 2006 |
| upper thermal niche* | 18.1 | wild | 41ºN | Santiago *et al.* 2016 |
| upper thermal niche* | 18.7 | wild | 41ºN | Santiago *et al.* 2016 |
| critical feeding temperature | 19.4 | laboratory | 54ºN | Elliott *et al.* 1995 |
| critical feeding temperature | ≥23 | laboratory | 59ºN | Forseth *et al.* 2009 |
| incipient lethal temperature* | 24.7 | laboratory | 54ºN | Elliott 1981 |
| ultimate | 27.8 | laboratory | Norway | Grande and Andersen 1991 |
| ultimate** | 29.7 | laboratory | 54ºN | Elliott 2000 |

*: seven days; **: 10 min.

**Table 4. Seasonal significant changes of flow variables in percentage (DJF: winter, MAM: spring, JJA: summer, SON: autumn) in H-2050 and H-2099, and RCP4.5 and RCP8.5 scenarios.**

### 2050 HORIZON

| scenario | variable | period | Tormes-Hoyos | Tormes-Barco | Cega-Pajares | Cega-Lastras | Pirón | Lozoya | Tagus | Gallo | Cabrillas | Ebron |
|---|---|---|---|---|---|---|---|---|---|---|---|---|
| RCP4.5 | Runoff Q=0 | annual | | | | | | -11.7 | -13.5 | | | |
| | Q=0 | annual | | | | | | | | | | |
| | Qmin | DJF | | | | -49.5 | | | | | | |
| | | MAM | | -56.7 | | -34.6 | | | | | | |
| | | JJA | | | | -39.7 | | -93.8 | | -28.0 | | |
| | | SON | | | -50.6 | -41.4 | | -106.0 | | | | |
| | Q10 | DJF | | | | -44.8 | | | | | | |
| | | MAM | | | | -25.4 | | | | | | |
| | | JJA | | | | -43.3 | | -92.9 | | -18.3 | | |
| | | SON | | | | -36.7 | | -98.1 | | | | |
| | Qmean | DJF | | | | | | | | | | |
| | | MAM | | | | -33.9 | | -7.9 | -22.0 | | | |
| | | JJA | | | | | 51.6 | -48.7 | | -13.0 | | |
| | | SON | -32.9 | | | | | -26.1 | | | -16.4 | |
| | Qmax | DJF | | | | | | | | | | |
| | | MAM | | | | -30.3 | | -30.3 | -28.7 | | | |
| | | JJA | | | | | | | | | | |
| | | SON | | | | | | | | | -34.7 | 19.6 |
| RCP8.5 | Runoff Q=0 | annual | | | | -13.6 | | -15.5 | -25.8 | -12.2 | -12.7 | |
| | Q=0 | annual | | | | 147.6 | | | | | | |
| | Qmin | DJF | | | | -32.9 | | | | | | |
| | | MAM | | | | -41.3 | | -39.4 | | | | -12.1 |
| | | JJA | -43.1 | | | | | -73.8 | | -33.9 | | |
| | | SON | -88.0 | | | -42.7 | | -49.7 | | | | |
| | Q10 | DJF | | | | -43.1 | | | -23.5 | | | -8.8 |
| | | MAM | | | | -33.1 | | -31.9 | -19.6 | | | |
| | | JJA | | | | -48.4 | | -89.1 | | -17.9 | | |
| | | SON | -40.6 | | | -45.2 | | -94.6 | | | | |
| | Qmean | DJF | | -16.6 | | | | | -30.3 | | -17.1 | |
| | | MAM | | | | | | -15.0 | -35.4 | | -9.7 | |
| | | JJA | | | | -44.0 | 28.6 | -47.2 | | -10.0 | -8.5 | |
| | | SON | | | | -40.6 | | -31.1 | | -14.5 | | |
| | Qmax | DJF | | | | | | -9.4 | | | | |
| | | MAM | | | | -33.6 | | | -30.9 | | | |
| | | JJA | | | | | | | -33.9 | | | |
| | | SON | | | -34.7 | -40.4 | | | | -22.3 | | |

### 2099 HORIZON

| scenario | variable | period | Tormes-Hoyos | Tormes-Barco | Cega-Pajares | Cega-Lastras | Pirón | Lozoya | Tagus | Gallo | Cabrillas | Ebron |
|---|---|---|---|---|---|---|---|---|---|---|---|---|
| RCP4.5 | Runoff Q=0 | annual | | -11.3 | | -8.6 | -17.0 | -11.5 | -12.9 | -6.5 | -9.1 | |
| | Q=0 | annual | | | | 85.8 | 42.7 | | | | | |
| | Qmin | DJF | | | | | | -9.3 | | 8.9 | | |
| | | MAM | | -28.8 | | -29.2 | | -22.7 | -14.8 | -13.0 | | -8.7 |
| | | JJA | | | | -9.9 | | -85.0 | | -18.0 | | |
| | | SON | | | | | -14.7 | -76.8 | | -11.5 | | |
| | Q10 | DJF | | | | -27.8 | | | | | | |
| | | MAM | -10.5 | -25.2 | | -19.4 | -23.3 | -19.2 | -17.2 | -8.2 | -11.6 | -6.2 |
| | | JJA | | | | -23.8 | | -48.6 | | -15.2 | -6.3 | |
| | | SON | | | | | -17.1 | -86.9 | | -11.0 | | |
| | Qmean | DJF | | | | | | | | | -11.6 | 6.8 |
| | | MAM | -10.5 | -15.1 | | -37.3 | -28.1 | -9.0 | -20.9 | -9.2 | | -5.3 |
| | | JJA | | | | | | -42.2 | -6.1 | -6.5 | -8.4 | 5.7 |
| | | SON | | | | | | -26.2 | | -7.1 | | |
| | Qmax | DJF | | | | | | | | | | |
| | | MAM | | | | -27.8 | -30.1 | -20.7 | -20.9 | -11.4 | -15.9 | |
| | | JJA | | | | | | -17.0 | -18.5 | -13.2 | | -8.7 |
| | | SON | | | | | | | | | | |
| RCP8.5 | Runoff Q=0 | annual | -7.7 | -12.5 | -12.5 | -37.7 | -48.6 | -39.0 | -32.1 | -19.0 | -17.5 | |
| | Q=0 | annual | | | -30.7 | 332.2 | 116.9 | | | 203.5 | | |
| | Qmin | DJF | | | | -58.3 | -69.4 | -41.8 | -29.3 | | | |
| | | MAM | -38.1 | -38.2 | | -69.5 | -31.2 | -68.9 | -32.2 | -11.6 | -18.3 | -12.0 |
| | | JJA | 66.2 | | | -53.3 | -14.2 | -111.8 | | -36.2 | -13.1 | |
| | | SON | 24.7 | | -24.0 | -51.4 | -16.0 | -93.8 | | -30.5 | | |
| | Q10 | DJF | -33.7 | -36.2 | | -62.9 | -62.7 | -37.0 | -37.1 | -12.4 | -16.2 | 14.8 |
| | | MAM | 67.9 | 46.8 | | -61.4 | -33.3 | -55.3 | -34.7 | -14.6 | -19.2 | |
| | | JJA | | | | -50.4 | | -72.0 | | -28.0 | -9.4 | -6.2 |
| | | SON | | | -12.9 | -53.2 | -18.9 | -108.0 | | -19.9 | -7.9 | |
| | Qmean | DJF | | -23.4 | | -40.0 | -55.3 | -21.1 | -40.4 | -20.3 | -19.2 | 5.8 |
| | | MAM | -22.5 | -21.4 | | -28.9 | -49.3 | -38.0 | -38.8 | -19.4 | -16.7 | -12.9 |
| | | JJA | 27.5 | 28.5 | | -65.7 | | -80.2 | | -17.0 | -11.2 | 12.6 |
| | | SON | | -22.0 | -37.4 | -51.5 | -50.6 | -59.6 | -21.5 | -17.5 | -18.0 | 4.4 |
| | Qmax | DJF | | -26.4 | | -20.5 | -44.3 | -14.4 | -32.5 | -32.5 | -27.2 | -11.1 |
| | | MAM | | -12.9 | | | -50.1 | -24.6 | -24.5 | -26.2 | -22.8 | |
| | | JJA | | | | -62.8 | -47.7 | -66.9 | -21.2 | -11.7 | | |
| | | SON | | -23.0 | -40.2 | -47.7 | | -38.1 | -45.7 | -24.3 | -24.6 | 26.2 |

**Table 5. Bayesian (BIC) and Akaike (AIC) information criteria values for the stream-temperature models with five and eight parameters.**

| Site | BIC 5 | BIC 8 | AIC 5 | AIC 8 |
|------|-------|-------|-------|-------|
| Tormes2 | 2075.7 | 1911.8 | 2108.5 | 1837.9 |
| Tormes3 | 2346.7 | 2274.2 | 2346.7 | 2234.8 |
| Cega1 | 1814.8 | 1731.9 | 1789.1 | 1693.4 |
| Pirón1 | 1725.9 | 1530.5 | 1700.2 | 1492.0 |
| Lozoya1 | 5097.3 | 4924.6 | 5065.2 | 4876.4 |
| Lozoya2 | 3979.3 | 3927.9 | 3948.6 | 3881.9 |
| Lozoya3 | 3841.6 | 3673.6 | 3811.5 | 3628.5 |
| Lozoya4 | 5076.6 | 4735.3 | 5044.8 | 4687.5 |
| Cabrillas | 2552.9 | 2172.2 | 2523.1 | 2127.4 |
| Ebrón | 624.8 | 169.8 | 598.5 | 130.3 |
| Vallanca1 | 1438.9 | 1359.9 | 1410.6 | 1317.3 |
| Vallanca2 | 1322.1 | 1279.7 | 1293.7 | 1237.1 |

**Table 6.** Parameter values of the stream temperature models for every thermograph site (λ is a dimensionless parameter) and values of the performance indicators Residual Standard Error (RSE) and Nash Sutcliffe Efficiency index (NSE).

| site | μ (ºC) | α (ºC) | α-μ (ºC) | γ (ºC⁻¹) | β (ºC) | λ | ω (ºC) | δ (m⁻³s) | τ (m³s⁻¹) | RSE (ºC) | NSE |
|---|---|---|---|---|---|---|---|---|---|---|---|
| Aravalle | 0.1 | 23.3 | 23.2 | 0.14 | 10.79 | -0.31 | | | | 2.31 | 0.70 |
| Barbellido | 1.1 | 19.2 | 18.1 | 0.23 | 12.19 | -0.30 | | | | 3.29 | 0.81 |
| Gredos Gorge | 2.4 | 19.0 | 16.6 | 0.18 | 14.06 | -0.29 | | | | 3.68 | 0.71 |
| Tormes1 | -1.1 | 20.8 | 21.9 | 0.16 | 11.00 | -0.35 | | | | 1.97 | 0.83 |
| Tormes2 | 3.4 | 24.6 | 21.3 | 0.14 | 11.82 | -0.31 | -3.98 | 61.43 | 0.37 | 1.56 | 0.78 |
| Tormes3 | -0.1 | 30.5 | 30.6 | 0.12 | 12.61 | -0.39 | -2.85 | 223.69 | 0.22 | 1.62 | 0.79 |
| Pirón1 | -1.4 | 19.1 | 20.5 | 0.09 | 14.33 | -0.18 | -2.08 | 72.82 | 0.15 | 0.97 | 0.91 |
| Pirón2 | 0.6 | 15.5 | 14.9 | 0.22 | 12.11 | -0.23 | | | | 3.69 | 0.82 |
| Pirón3 | 7.2 | 14.0 | 6.8 | 0.29 | 10.17 | -0.10 | | | | 1.16 | 0.77 |
| Pirón4 | -0.6 | 18.3 | 18.9 | 0.15 | 8.20 | -0.29 | | | | 1.26 | 0.85 |
| Pirón5 | -4.6 | 21.7 | 26.3 | 0.13 | 8.92 | -0.40 | | | | 1.57 | 0.39 |
| Cega1 | 1.4 | 15.6 | 14.2 | 0.19 | 15.92 | -0.22 | -1.53 | 112.98 | 0.27 | 1.17 | 0.88 |
| Cega2 | -0.6 | 18.0 | 18.6 | 0.17 | 16.03 | -0.31 | | | | 1.81 | 0.85 |
| Cega3 | -2.0 | 24.7 | 26.6 | 0.14 | 15.35 | -0.41 | | | | 2.83 | 0.85 |
| Cega4 | -0.3 | 19.9 | 20.2 | 0.16 | 12.21 | -0.35 | | | | 1.65 | 0.87 |
| Cega5 | -2.4 | 18.1 | 20.5 | 0.13 | 7.85 | -0.31 | | | | 0.87 | 0.84 |
| Cega6 | 0.7 | 22.4 | 21.7 | 0.13 | 13.84 | -0.38 | | | | 2.34 | 0.79 |
| Lozoya1 | 0.4 | 19.5 | 19.1 | 0.18 | 11.90 | -0.24 | -1.33 | 11.93 | 0.41 | 1.13 | 0.90 |
| Lozoya2 | 0.3 | 20.2 | 20.0 | 0.19 | 11.63 | -0.28 | -1.27 | 13.16 | 0.38 | 1.17 | 0.90 |
| Lozoya3 | 1.1 | 21.0 | 19.9 | 0.19 | 10.62 | -0.29 | -1.74 | 23.44 | 0.49 | 1.23 | 0.89 |
| Lozoya4 | 1.7 | 22.0 | 20.2 | 0.17 | 10.29 | -0.27 | -2.19 | 17.04 | 0.48 | 1.17 | 0.90 |
| Tagus-Peralejos | 1.1 | 21.0 | 19.9 | 0.11 | 11.01 | -0.17 | | | | 1.00 | 0.89 |
| Tagus-Poveda | 1.3 | 20.4 | 19.2 | 0.15 | 9.95 | -0.38 | | | | 1.26 | 0.83 |
| Gallo | 0.4 | 20.2 | 19.8 | 0.13 | 7.76 | -0.18 | | | | 0.90 | 0.92 |
| Cabrillas | 8.3 | 15.3 | 7.0 | 0.21 | 9.23 | -0.04 | -1.38 | 13.56 | 1.25 | 1.65 | 0.90 |
| Ebrón | 5.5 | 17.0 | 11.4 | 0.07 | 6.58 | -0.06 | 1.73 | -1.78 | 3.16 | 0.27 | 0.86 |
| Vallanca1 | -0.5 | 16.9 | 17.4 | 0.09 | 4.70 | -0.12 | 1.95 | -5.21 | 4.85 | 0.53 | 0.88 |
| Vallanca2 | 1.4 | 16.8 | 15.4 | 0.10 | 5.36 | -0.11 | 1.54 | -11.29 | 5.29 | 0.50 | 0.89 |
| Palancia1 | 11.7 | 15.3 | 3.6 | 0.19 | 13.92 | -0.03 | | | | 0.23 | 0.93 |
| Palancia2 | 9.3 | 16.1 | 6.8 | 0.27 | 12.73 | -0.11 | | | | 0.59 | 0.88 |
| Villahermosa | 7.8 | 18.0 | 10.2 | 0.27 | 16.46 | -0.20 | | | | 1.02 | 0.85 |

**Table 7. Maximum daily mean stream temperature (°C) in each site in the year 2015 and the horizons H-2050 and H-2099. Both scenarios (RCP4.5 and RCP8.5) are showed. Bold numbers: values > 18.7°C.**

| | Maximum daily mean stream temperature (ºC) | | | | | |
| | RCP4.5 | | | RCP8.5 | | |
| site | 2015 | H-2050 | H-2099 | 2015 | H-2050 | H-2099 |
|---|---|---|---|---|---|---|
| **Aravalle** | **19.8** | **20.4** | **21.0** | **19.8** | **20.7** | **22.0** |
| **Barbellido** | 17.9 | 18.4 | **18.7** | 17.9 | 18.6 | **19.3** |
| Gredos Gorge | 16.5 | 17.1 | 17.6 | 16.5 | 17.4 | 18.5 |
| **Tormes1** | 18.1 | 18.6 | **19.1** | 18.0 | **18.9** | **20.1** |
| **Tormes2** | **20.5** | **21.2** | **21.4** | **20.7** | **21.1** | **22.1** |
| **Tormes3** | **21.8** | **22.7** | **23.1** | **22.4** | **22.4** | **24.5** |
| Cega1 | 12.4 | 13.1 | 13.6 | 12.5 | 13.3 | 14.0 |
| Cega2 | 15.2 | 15.9 | 16.3 | 15.2 | 16.1 | 17.3 |
| **Cega3** | **19.8** | **20.7** | **21.4** | **19.8** | **21.0** | **22.8** |
| **Cega4** | 18.1 | 18.5 | **18.9** | 18.1 | **18.7** | **19.7** |
| Cega5 | 16.6 | 16.9 | 17.3 | 16.6 | 17.1 | 17.8 |
| **Cega6** | **18.7** | **19.5** | **19.9** | **18.8** | **19.6** | **21.0** |
| Pirón1 | 12.9 | 13.8 | 14.2 | 13.2 | 13.9 | 15.6 |
| Pirón2 | 14.9 | 15.1 | 15.4 | 14.9 | 15.3 | 15.7 |
| Pirón3 | 14.1 | 14.1 | 14.2 | 14.0 | 14.2 | 14.3 |
| Pirón4 | 17.2 | 17.5 | 17.8 | 17.2 | 17.7 | 18.3 |
| **Pirón5** | **19.3** | **19.8** | **20.2** | **19.3** | **20.0** | **21.1** |
| **Lozoya1** | 16.8 | 17.4 | 17.8 | 16.8 | 17.6 | **18.8** |
| **Lozoya2** | 17.6 | 18.1 | 18.6 | 17.5 | 18.4 | **19.6** |
| **Lozoya3** | **19.0** | **19.5** | **19.9** | **18.9** | **19.7** | **20.8** |
| **Lozoya4** | **19.5** | **20.0** | **20.5** | **19.5** | **20.3** | **21.4** |
| Tagus-Peralejos | 16.7 | 17.2 | 17.6 | 16.6 | 17.4 | 18.6 |
| **Tagus-Poveda** | 18.1 | 18.6 | **19.0** | 18.1 | **18.8** | **19.9** |
| **Gallo** | 17.9 | 18.3 | 18.6 | 17.9 | 18.4 | **19.3** |
| Cabrillas | 14.9 | 15.0 | 15.1 | 14.9 | 15.1 | 15.2 |
| Ebrón | 16.2 | 16.5 | 16.5 | 16.2 | 16.5 | 17.0 |
| Vallanca1 | 16.8 | 17.1 | 17.3 | 16.8 | 17.2 | 17.9 |
| Vallanca2 | 16.5 | 16.8 | 17.0 | 16.5 | 16.9 | 17.5 |
| Palancia1 | 15.0 | 15.1 | 15.1 | 15.0 | 15.1 | 15.3 |
| Palancia2 | 16.0 | 16.1 | 16.1 | 16.0 | 16.1 | 16.4 |
| Vistahermosa | 16.0 | 16.1 | 16.1 | 16.0 | 16.1 | 16.5 |