# Peer review of "Waning habitats due to climate change: the effects of changes in streamflow and temperature at the rear edge of the distribution of a cold-water fish"

_Hydrology and Earth System Sciences, 2016_

## Referee Comment (RC1) · V. Ouellet (Referee) · 6 Feb 2017

General comments ć Since both air and water temperature terms are used in the paper, please specify throughout the manuscript to which term the authors are referring, thus avoiding the use of only the term temperature since in some paragraph it could be confusing. ć In the IS notation, there is a non-breaking space between numbers and oC. Please modify throughout the document. ć Those are difficult results to present but the presentation could be improved (see specific comments) to help the reader having a better understanding and be able to have a quantitative appreciation
of the differences between scenarios.

Specific comments âAc P2-L2: physiological functions such as blood... can you be more specific? Are you referring to the blood cell formation/maturation? aĂć P2-L13: add by between ecosystems and altering ć P2-L15: will be interesting to add with the geographical location a mean increase value... ć P2-L32: is instead of was. ć P3-L10: I will suggest merging the two sentences, directly mentioning changes in fish habitat suitability and availability. ć P4-L19: what do you mean by not probable? aĂć P5: were the logger shaded and tested prior to deployment? Did you check if the data from AEMET were corrected for change in instruments or station location trough time? ć P9-L19: A table summarizing the different values found across different geographical range will be interesting here. The 7 days period is usually used for incipient lethal temperature (ILT) (it is highly variable depending on acclimation and the rate of change in water temperatures) and the values are higher than the one chose in this study. Studies on thermal tolerances usually use shorter exposure time... I feel more explanation is needed to understand if the goal is to assess the changes regarding to ILT so brown trout will be expected to disappear from the habitat or regarding to suitable thermal tolerances linked to growth and other physiological parameters (as the chosen threshold suggest), which implies that the specie may still be found but not be performing. I think the manuscript will benefit from a slightly extended justification. aÅć P14-Figure 6: this figure is difficult to read, text overlap, difficulty to discern the white dots, etc. I am not sure which sites belong to which clusters from the figures. May be split in 2 figures based on RPC4.5 and 8.5? ć P16-Figure 8: This figure is also hard to read. May be have different temperature ranges for the 2 scenarios so the results for RCP 4.5 are easier to read. aAć P18: a table or figure with the water temperature reached (to present not only the consecutive days above the threshold but also by how much this threshold is passed) will give a deeper understanding of the consequences for thermal habitat and strengthen the discussion. ć P19-L10: I will suggest use detailed prediction resolution instead of finer (or another synonym). ć P20-L20: This does not guaranty model robustness... You should present model performance results
or at least explain how you tested the model robustness or change this paragraph. ć P22-L8: do you mean maturation or development instead of their duration?

---

## Author Comment (AC1) · 3 Mar 2017

Dear Dr. Ouellet, We appreciate very much the meticulous work that you did. Your comments are very valuable for us and it is doubtless our work will enrich from them.

GENERAL COMMENTS:

* "Since both air and water temperature terms are used in the paper, please specify throughout the manuscript to which term the authors are referring, thus avoiding the use of only the term temperature since in some paragraph it could be confusing." Answer: Thank you. Truly, many times the context does not clear up of which temperature

we are talking about.

\* "In the IS notation, there is a non-breaking space between numbers and °C. Please modify throughout the document." Answer: Certainly, we should have been more careful to follow the same criteria throughout the manuscript.

\* "Those are difficult results to present but the presentation could be improved (see specific comments) to help the reader having a better understanding and be able to have a quantitative appreciation of the differences between scenarios." Answer: Taken into account.

SPECIFIC COMMENTS: P2-L2: "physiological functions such as blood. . . can you be more specific? Are you referring to the blood cell formation/maturation? We refer to the blood physiological function." Answer: Certainly, this must be clarified this. A more appropriate way to say it could be: "physiological functions such as blood function. . ." We will change it.

P2-L13: "add by between ecosystems and altering." Answer: Thank you for noticing.

P2-L15: "will be interesting to add with the geographical location a mean increase value."Answer: We changed the wording to: "Stream temperature increases have been documented for the last decades throughout the globe, in Europe (e.g., Orr et al., 2015, reported a mean stream temperature average increase by 0.03°C per year in England and Wales), Asia (e.g., Chen et al., 2016, mean stream temperature increase by 0.029-0.046°C per year at Yongan River; eastern China), America (e.g., Kaushal et al., 2010, mean stream temperature increases by 0.009–0.077°C per year) and Australia (e.g., Chessman, 2009, stream temperature increases by 0.12°C per year between sampling campaigns)."

P2-L32: "is instead of was." Answer: Thank you. It was changed.

P3-L10: "I will suggest merging the two sentences, directly mentioning changes in fish habitat suitability and availability." Answer: New wording: "The results are daily values

to be used for the assessment of fish habitat suitability and availability."

P4-L19: "what do you mean by not probable?" Answer: Territorial planning does not consider significant changes of land-use at mid-century and, objectively, changes are not expected after that horizon because a high percentage of the territory is protected. (This will be included in the corrected manuscript.)

P5: "were the logger shaded and tested prior to deployment? Did you check if the data from AEMET were corrected for change in instruments or station location trough time?" Answer: Loggers were tested for malfunction before been deployed and they were placed avoiding direct sunshine. Air temperature and precipitation data obtained from AEMET were tested to assess their reliability by applying a homogeneity test. This test is based on a two-sample Kolmogorov–Smirnov test, and it marks years as possibly inhomogeneous data. In a second phase, the marked years are matched against the distribution of the entire series to determine if they have true inhomogeneities, searching for possible dissimilarities between the empirical distribution functions. This technique was used by us in the previous paper: Santiago et al. (2016). Only reliable series were used. The location of the stations did not change in the studied period. These explanations will be included in the manuscript.

P9-L19: "A table summarizing the different values found across different geographical range will be interesting here. The 7 days period is usually used for incipient lethal temperature (ILT) (it is highly variable depending on acclimation and the rate of change in water temperatures) and the values are higher than the one chose in this study. Studies on thermal tolerances usually use shorter exposure time... I feel more explanation is needed to understand if the goal is to assess the changes regarding to ILT so brown trout will be expected to disappear from the habitat or regarding to suitable thermal tolerances linked to growth and other physiological parameters (as the chosen threshold suggest), which implies that the specie may still be found but not be performing. I think the manuscript will benefit from a slightly extended justification." Answer: The new table (new Table 3) is in attached file. We don't talk about thermal tolerance, we

want to talk about realized thermal niche and on the conditions in which the exclusion probability is high for trout. The realized niche must reflect the energetic efficiency: long time above that threshold makes the animals less efficient competitors and its performance would decrease critically (Magnuson et al. 1979, Verberk et al, 2016). Thus, we focus our study on realized thermal niche. In experiments in which water modelling was done, it was usual to use weekly moving average stream temperature and to contrast it against a threshold, like the one given by Elliott et al. (1995). On the other hand, the usual time for determining thermal tolerance is 7 consecutive days (Elliott and Elliott 2010). However, using the weekly moving average could introduce errors such as the overestimation of the importance of a threshold. This is because a given weekly moving average does not indicate that every considered daily average is equal to or higher than the weekly moving average. Furthermore, in Santiago et al. (2016), we tested the adequacy of using: (1) daily mean stream temperature (DM); (2) 7-day moving average of DM; (3) daily maximum stream temperature (DMax); and (4) 7-day moving average of DMax to model thermal behaviour of streams and to determine the brown trout presence/absence ecological thresholds. We found that DM was the best solution to model thermal behaviour of the streams, and the study of events of 7 consecutive days above the threshold was better than 7-day moving average. In addition, the used threshold (18.7°C during 7 -or more- consecutive days) was originally determined in one of the streams of this paper (Cega stream). Consequently, daily mean temperature and 7 consecutive days threshold were used in this study because they better reflect the average conditions that trout experience for an extended period.

* Bustillo, V., Moatar, F., Ducharne, A., Thiéry, D., & Poirel, A. (2013). A multimodel comparison for assessing water temperatures under changing climate conditions via the equilibrium temperature concept: case study of the Middle Loire River, France. Hydrological Processes. Retrieved from http://onlinelibrary.wiley.com/doi/10.1002/hyp.9683/full * Edinger, J. E., Duttweiler, D. W., & Geyer, J. C. (1968). The response of water temperatures to meteorological conditions. Water Resources Research, 4(5), 1137–1143. * Elliott, J., & Elliott, J. (2010).
Temperature requirements of Atlantic salmon Salmo salar, brown trout Salmo trutta and Arctic charr Salvelinus alpinus: predicting the effects of climate change. Journal of Fish Biology, 77(8), 1793–1817. https://doi.org/10.1111/j.1095-8649.2010.02762.x * Elliott, J. M., Hurley, M. A., & Fryer, J. (1995). A new, improved growth model for brown trout, Salmo trutta. Functional Ecology, 9(2), 290–298. * Magnuson, J. J., Crowder, L. B., & Medvick, P. A. (1979). Temperature as an Ecological Resource. American Zoologist, 19(1), 331–343. https://doi.org/10.1093/icb/19.1.331 * Santiago, J. M., García de Jalón, D., Alonso, C., Solana, J., Ribalaygua, J., Pórtoles, J., & Monjo, R. (2016). Brown trout thermal niche and climate change: expected changes in the distribution of cold-water fish in central Spain. Ecohydrology, 9(3), 514–528. https://doi.org/10.1002/eco.1653 * Verberk, W. C. E. P., Durance, I., Vaughan, I. P., & Ormerod, S. J. (2016). Field and laboratory studies reveal interacting effects of stream oxygenation and warming on aquatic ectotherms. Global Change Biology, 22(5), 1769–1778. https://doi.org/10.1111/gcb.13240

P14-Figure 6: "this figure is difficult to read, text overlap, difficulty to discern the white dots, etc. I am not sure which sites belong to which clusters from the figures. May be split in 2 figures based on RPC4.5 and 8.5?" Answer: We have tried several alternatives (even using GIS-maps) and, finally, we selected the attached solution as optimal.

P16-Figure 8: "This figure is also hard to read. May be have different temperature ranges for the 2 scenarios so the results for RCP 4.5 are easier to read." Answer: I understand your concerns but still we think that, to compare both scenarios, keeping the same scale makes it easier to see the differences between them.

P18: "a table or figure with the water temperature reached (to present not only the consecutive days above the threshold but also by how much this threshold is passed) will give a deeper understanding of the consequences for thermal habitat and strengthen the discussion." Answer: Please, see attached file.

P19-L10: "I will suggest use detailed prediction resolution instead of finer (or another

synonym)." Answer: Yes. We'll do it.

P20-L20: "This does not guaranty model robustness... You should present model performance or at least explain how you tested the model robustness or change this paragraph." Answer: Certainly, the wording of the sentence was not good. We changed it to: "We used a regression-based method to assess the impact of climate change in river temperatures. Bustillo et al. (2013) recommended this type of methods that rely on logistic approximations of equilibrium temperatures (Edinger et al., 1968), which are at least as robust as the most refined classical heat balance models."

P22-L8: "do you mean maturation or development instead of their duration?" Answer: Yes, "development" is better.

――――――――――――――――――

*Table 3. Different classes of thermal thresholds for emerged trout classes found in literature. Type of experiments differentiate experiments with controlled (laboratory) and uncontrolled (wild) temperature. Latitude of the experiments' location is showed.*

| variable | temperature (°C) | type of experiment | latitude | reference |
|---|---|---|---|---|
| maximum growth | 13.1 | laboratory | 54°N | Elliott et al. 1995 |
| maximum growth | 16 | laboratory | 61°N | Forseth & Jonsson 1994 |
| maximum growth | 16.9 | laboratory | 43°N | Ojanguren et al. 2001 |
| maximum growth | 13.2 | wild | 43°N | Lobón-Cerviá & Rincón 1998 |
| maximum growth | 13 | wild | 41°S | Allen 1985 |
| maximum growth | 15.4-19.1 | laboratory | 59°N | Forseth et al. 2009 |
| thermal optimum | 14.2 | wild | 47°N | Hari et al. 2006 |
| upper growth limit | 19.5 | wild | 41°S | Allen 1985 |
| upper thermal niche | 20 | wild | 47°N | Hari et al. 2006 |
| upper thermal niche* | 18.1 | wild | 41°N | Santiago et al. 2016 |
| upper thermal niche* | 18.7 | wild | 41°N | Santiago et al. 2016 |
| critical feeding temperature | 19.4 | laboratory | 54°N | Elliott et al. 1995 |
| critical feeding temperature | ≥23 | laboratory | 59°N | Forseth et al. 2009 |
| incipient lethal temperature* | 24.7 | laboratory | 54°N | Elliott 1981 |
| ultimate | 27.8 | laboratory | Norway | Grande & Andersen 1991 |
| ultimate** | 29.7 | laboratory | 54°N | Elliott 2000 |

*: 7 days; **: 10 min.

**Fig. 1.**

[Figure]

Figure 6. Study sites clustered by the predicted change ratios of the monthly mean streamflow (gauging stations) and by the predicted increase of the monthly mean temperature (ºC, at water temperature recording sites) at H-2050 and H-2099 for the RCP4.5 and RCP8.5 scenarios. Axes show geographic position (UTM coordinates). Colours and numbers show clusters.

[Figure]

**Fig. 2.**

*Table 7. Maximum daily mean stream temperature (°C) at each site at the current time (2015) and horizons H2050 and H2099.Both scenarios (RCP4.5 and 8.5) are showed.*

| | maximum daily mean stream temperature (°C) | | | | | |
| | | RCP4.5 | | | RCP8.5 | |
| site | 2015 | H2050 | H2099 | 2015 | H2050 | H2099 |
|---|---|---|---|---|---|---|
| Aravalle | 19.8 | 20.4 | 21.0 | 19.8 | 20.7 | 22.0 |
| Barbellido | 17.9 | 18.4 | 18.7 | 17.9 | 18.6 | 19.3 |
| Gredos Gorge | 16.5 | 17.1 | 17.6 | 16.5 | 17.4 | 18.5 |
| Tormes1 | 18.1 | 18.6 | 19.1 | 18.0 | 18.9 | 20.1 |
| Tormes2 | 20.5 | 21.2 | 21.4 | 20.7 | 21.1 | 22.1 |
| Tormes3 | 21.8 | 22.7 | 23.1 | 22.4 | 22.4 | 24.5 |
| Cega1 | 12.4 | 13.1 | 13.6 | 12.5 | 13.3 | 14.0 |
| Cega2 | 15.2 | 15.9 | 16.3 | 15.2 | 16.1 | 17.3 |
| Cega3 | 19.8 | 20.7 | 21.4 | 19.8 | 21.0 | 22.8 |
| Cega4 | 18.1 | 18.5 | 18.9 | 18.1 | 18.7 | 19.7 |
| Cega5 | 16.6 | 16.9 | 17.3 | 16.6 | 17.1 | 17.8 |
| Cega6 | 18.7 | 19.5 | 19.9 | 18.8 | 19.6 | 21.0 |
| Pirón1 | 12.9 | 13.8 | 14.2 | 13.2 | 13.9 | 15.6 |
| Pirón2 | 14.9 | 15.1 | 15.4 | 14.9 | 15.3 | 15.7 |
| Pirón3 | 14.1 | 14.1 | 14.2 | 14.0 | 14.2 | 14.3 |
| Pirón4 | 17.2 | 17.5 | 17.8 | 17.2 | 17.7 | 18.3 |
| Pirón5 | 19.3 | 19.8 | 20.2 | 19.3 | 20.0 | 21.1 |
| Lozoya1 | 16.8 | 17.4 | 17.8 | 16.8 | 17.6 | 18.8 |
| Lozoya2 | 17.6 | 18.1 | 18.6 | 17.5 | 18.4 | 19.6 |
| Lozoya3 | 19.0 | 19.5 | 19.9 | 18.9 | 19.7 | 20.8 |
| Lozoya4 | 19.5 | 20.0 | 20.5 | 19.5 | 20.3 | 21.4 |
| TagusPeralejos | 16.7 | 17.2 | 17.6 | 16.6 | 17.4 | 18.6 |
| TagusPoveda | 18.1 | 18.6 | 19.0 | 18.1 | 18.8 | 19.9 |
| Gallo | 17.9 | 18.3 | 18.6 | 17.9 | 18.4 | 19.3 |
| Cabrillas | 14.9 | 15.0 | 15.1 | 14.9 | 15.1 | 15.2 |
| Ebrón | 16.2 | 16.5 | 16.5 | 16.2 | 16.5 | 17.0 |
| Vallanca1 | 16.8 | 17.1 | 17.3 | 16.8 | 17.2 | 17.9 |
| Vallanca2 | 16.5 | 16.8 | 17.0 | 16.5 | 16.9 | 17.5 |
| Palancia1 | 15.0 | 15.1 | 15.1 | 15.0 | 15.1 | 15.3 |
| Palancia2 | 16.0 | 16.1 | 16.1 | 16.0 | 16.1 | 16.4 |
| Vistahermosa | 16.0 | 16.1 | 16.1 | 16.0 | 16.1 | 16.5 |

**Fig. 3.**

---

## Referee Comment (RC2) · Anonymous Referee #2 · 22 Mar 2017

The authors present modelling study to evaluate effects of changing climate on streamflow and temperature changes in several rivers in Spain. They further analyze effect of change to habitat changes of a cold-water fishes. The topic is interesting, up-to-date and with global interest. Authors use suitable approaches and analysis techniques, and manuscript has some potential to be published in high quality journal. However, manuscript contains weakness which needs to handle before acceptance, mainly related to writing style and language.

Main comments.

The paper is long and confusing with at times repeat and/or not needed information. It needs to be re-written and streamlined for clarity. Please separate Discussion and Conclusions into two different sections. Many sections are difficult to read and follow, especially methods, results and discussion need to be re-written. Try to make manuscript more compact and avoid repeating. Use of several abbreviation and various terminologies makes "story" in the manuscript sometimes difficult to follow and understand the main points. Manuscript contains lots of results, but authors should focus only to the main results. Is all small details needed to highlight? Language should be checked by native speaker.

Detailed comments:

- Please provide more specified objectives

- There has been newer IPCC climate scenarios (IPCC6). Please let readers know how this reflects to your results.

- Study sites: Please specify which kind of forest and geology sites contains

- Data collection: what are time periods for temperature data collection? How logger was installed? Was discharge measured from all sites?

- Hydrological modelling: Whole section is confusing, please clarify and make in more compact. Did authors calibrate M5 models with measured discharge from all sites? Was model validation done?

- Stream temperature modelling: Please re-write whole section

- Page 9, lines 10-14: Correct place for geology part? What geology classes where used?

- Page 10, lines 2-6: Please tell in more details how DEM was used to study stream continuum. Was this information mentioned in Results?

- Results: Tell first main results (in beginning of the paragraph). Please re-write results,

now they are difficult to follow.

- Figure 6: Not sure is this figure needed. At least need more explanation from main points.

- Figure 7: please tell geological classes already in methods

- Page 17: is all numerical results necessary to include to the text? Especially section 3.3.4 is challenging to read.

- Discussion: Please re-formulate and re-write. No detailed comments provided.

---

## Author Comment (AC2) · 4 Apr 2017

Thank you for your revision. We have read your comments carefully and proceeded to make the necessary changes. We have reviewing the manuscript to do the reading more fluid and to avoid redundancies, especially between text and figures/tables. We agree that abbreviations and terminology may make it complex to easily follow the text, however we strongly believe that they are essential for accuracy of the exposition. Nevertheless, we have reviewed the wording in order to make understanding easier. We are not native English speakers, therefore, we got the manuscript edited for proper

[Figure]

English language, grammar, punctuation, spelling, and overall style by native specialists. Please, see the attached certificate. However, we have given another "turn of the screw" to the whole manuscript.

Detailed comments: - Comment: "Please provide more specified objectives." Response: Done. We have included this paragraph to address this comment: Specifically, in this paper: (i) we assess both the streamflow and geology effects on stream temperature; (ii) we predict the changes in streamflow and stream temperature in the IPCC5 climate change scenarios; and (iii) we assess the expected effects of these changes on trout habitat aptitude.

- Comment: "There has been newer IPCC climate scenarios (IPCC6). Please let readers know how this reflects to your results." Response: As far as we can see today, we don't know any published results on IPCC6. CMIP6 initiative is in progress and experiments defined: • CMIP6 experimental design finalized • Forcing datasets for DECK and CMIP6 historical simulations finalized Thus, it is not possible for us to interpret how the new experimental designs affect our results. (Eyring, V., Bony, S., Meehl, G. A., Senior, C. A., Stevens, B., Stouffer, R. J., and Taylor, K. E.: Overview of the Coupled Model Intercomparison Project Phase 6 (CMIP6) experimental design and organization, Geosci. Model Dev., 9, 1937-1958, doi: 10.5194/gmd-9-1937-2016, 2016.)

- Comment: "Study sites: Please specify which kind of forest and geology sites contains." Response: We have taken into account this comment. Forest are mainly composed by coniferous belonging to genus Pinus (P. sylvestris, P. nigra, P. pinea, P. pinaster). This is specified in the new version. Sites geology is described in the main text and Table 1 (see also Figure 2) and in p 4 line7. Our concern on geology relies on its hydrological response. In the Iberian Peninsula are distinguished four main lithological classes: igneous, carbonated, detrital and volcanic. These main classes are used because resume very well their different behaviour relating to the water cycle.

- Comment: "Data collection: what are time periods for temperature data collection? How logger was installed? Was discharge measured from all sites?" Response: We have incorporated in the text additional information on these matters. Regarding time periods, they are specified in Table 1 (total recording days at each site). Loggers were kept recording all throughout the year. Also, loggers were tested for malfunction before being deployed, and they were placed avoiding direct sunshine (as specified in the response to the Referee 1). Discharge was obtained from 10 gauging stations of the official network (p 5 line 9, and Table 2). These 10 stations were used to model running flows.

- Comment: "Hydrological modelling: Whole section is confusing, please clarify and make in more compact. Did authors calibrate M5 models with measured discharge from all sites? Was model validation done?" Response: Done. Sites in which discharge was measured were the gauging station sites. Five-fold cross validation was done in each case.

- Comment: "Stream temperature modelling: Please re-write whole section." Response: Done. We agree that this section can be particularly complex but this complexity is somewhat inherent to the matter we address and we cannot see how it can be further simplified.

- Comment: "Page 9, lines 10-14: Correct place for geology part? What geology classes where used?" Response: We think this is a suitable place because geology is used to analyse particularities of the temperature models. As said above, geology (lithology) classes were described in page 4 line 7 (and following) and Table 1 (see also Figure 2).

- Comment: "Page 10, lines 2-6: Please tell in more details how DEM was used to study stream continuum. Was this information mentioned in Results?" Response: We have explained it better. An altitudinal interpolation of the parameters of the stream temperature models was performed and a digital elevation model (DEM, at a 5-m resolution,

obtained from LIDAR, IGN [National Geographic Institute of the Spanish Government]) was used to determine the geographic coordinates and the altitude (x, y, z) of the points at which the established threshold will be transgressed in the simulations of the effects of climate change. This results will make possible to determine the altitude and the percentage of the length of stream in which the suitable thermal conditions for the trout will be lost. The results of it are reflected in the cited usable length reductions (p 17 line 30 . . .56%, 11%, 66%...). This information has been also completed with altitudinal data.

- Comment: "Results: Tell first main results (in beginning of the paragraph). Please re-write results, now they are difficult to follow." Response: Done. The whole section has been rewritten.

- Comment: "Figure 6: Not sure is this figure needed. At least need more explanation from main points." Response: We think this is important for understanding the hydrological and thermal response at a glance. We have improved and simplified the explanation.

- Comment: "Figure 7: please tell geological classes already in methods." Response: It was done. Please, see page 4 line 7 (and following) and Table 1 (see also Figure 2).

- Comment: "Page 17: is all numerical results necessary to include to the text? Especially section 3.3.4 is challenging to read." Response: We completely agree. The section was too wordy. This have been mended by removing unnecessary descriptions of the results.

- Comment: "Discussion: Please re-formulate and re-write. No detailed comments provided." Response: Done.

Please also note the supplement to this comment:
http://www.hydrol-earth-syst-sci-discuss.net/hess-2016-606/hess-2016-606-AC2-supplement.pdf

[Figure]

**Supplement:**

**AMERICAN JOURNAL EXPERTS**

**EDITORIAL CERTIFICATE**

This document certifies that the manuscript listed below was edited for proper English language, grammar, punctuation, spelling, and overall style by one or more of the highly qualified native English speaking editors at American Journal Experts.

**Manuscript title:**

HOW MUCH FISH HABITAT WILL BE LOST DUE TO CLIMATE CHANGE? QUANTIFYING THE EFFECT OF STREAM FLOW AND TEMPERATURE CHANGES AT THE REAR EDGE OF THE BROWN TROUT NATURAL DISTRIBUTION

**Authors:**

José M. Santiago, Rafael Muñoz-Mas, Joaquín Solana, Diego García de Jalón, Carlos Alonso, Francisco Martínez-Capel, Javier Pórtoles, Robert Monjo, Jaime Ribalaygua

**Date Issued:**

**Certificate Verification Key:**

628B-CF88-BF3C-D483-B53F

[Figure]

This certificate may be verified at www.aje.com/certificate. This document certifies that the manuscript listed above was edited for proper English language, grammar, punctuation, spelling, and overall style by one or more of the highly qualified native English speaking editors at American Journal Experts. Neither the research content nor the authors' intentions were altered in any way during the editing process. Documents receiving this certification should be English-ready for publication; however, the author has the ability to accept or reject our suggestions and changes. To verify the final AJE edited version, please visit our verification page. If you have any questions or concerns about this edited document, please contact American Journal Experts at support@aje.com.

American Journal Experts provides a range of editing, translation and manuscript services for researchers and publishers around the world. Our top-quality PhD editors are all native English speakers from America's top universities. Our editors come from nearly every research field and possess the highest qualifications to edit research manuscripts written by non-native English speakers. For more information about our company, services and partner discounts, please visit www.aje.com.

---

## Referee Report (RR1)

Review comments for:

**Waning habitats due to climate change: the effects of changes in streamflow and temperature at the rear edge of the distribution of a cold-water fish**

**By:** José M. Santiago, Rafael Muñoz-Mas, Joaquín Solana, Diego García de Jalón, Carlos Alonso, Francisco Martínez-Capel, Javier Pórtoles, Robert Monjo, Jaime Ribalaygua

The subject paper represents a considerable investment in time and resources to assess the impact of climate change on the thermal habitat of a cold-water fish, the brown trout (*Salmo trutta*), in central Spain. It is an ambitious effort in collecting and organizing meteorologic, hydrologic and stream temperature data. The downscaling and bias-correction methods of the simulations from general circulation models are of satisfactory quality.

Where the work disappoints is in the methods used to develop state estimates of stream flow and water temperature.  The artificial intelligence-based method, M5, used to simulate stream flows, is from a 27-year old paper, not readily available for study by reviewers. The description of the method (Pages 7-8) is replete with jargon and very difficult to understand.  However, there is little reason to doubt that the authors applied the method incorrectly. In addition, it may well be the case that the results are within the bands of uncertainty that might be expected by applying a more modern hydrologic model.  So, while the paper does not make a compelling case for using this method rather than a more modern one, the results are probably adequate for the specific scientific question posed here.

The stream temperature modeling is even less compelling. The regression-based methodology ( Eq. 1) is *ad hoc* and one that has been criticized for its lack of ability to project the effects of climate (Arismendi et al, 2014). The authors incorrectly cite the

work of Piccolroaz et al, 2016 in support of their method. Rather than supporting regression methods like Eq. 1 of this paper, Piccolroaz et al, 2016, conclude that "Conversely, performances of purely regression-based or stochastic models are lower" than their model. It is a well-documented finding, however, that stream temperature is highly correlated with air temperature and, as is the case for the hydrologic model, M5, the results are likely to be within the uncertainty bands that would result from the application of one of the myriad models based on the thermal energy budget.

The development and analysis of this data set is noteworthy and worth publishing because of its environmental relevance. The outcomes from analyzing good, large data sets of stream temperature, hydrology and climate are reasonably robust in terms of the type of model being used. Based on this notion, it would seem the conclusions are also reasonable and, hopefully, of use to water resource planners.

The document also has some shortcomings in terms of an editorial nature, however, and would be improved in the following way:

* Have someone proofread it carefully.

* There are too few statistical measures of outcomes, particularly for water temperature.

* Use a term other than "rear edge" to define the upper range of satisfactory temperatures for brown trout.

* Explain "future running flows" or use a different term

* Define "agglomerative coefficient"

* Try to use fewer acronyms.

*Table S1 is not referenced in the main document and needs a much better description of what's in it.

* Check the references to make certain they are complete.

**References**

Arismendi, I., Safeeq, M., Dunham, J.B. and Johnson, S.L.: Can air temperature be used to project influences of climate change on stream temperature? Environ. Res. Lett., 9(8), 084015. doi: 10.1088/1748-9326/9/8/084015, 2014.

Piccolroaz, S., Calamita, E., Majone, B., Gallice, A., Siviglia, A. and Toffolon, M.: Prediction of river water temperature: a comparison between a new family of hybrid models and statistical approaches: Prediction of River Water Temperature. Hydrol. Process., 30(21), 3901–3917. https://doi.org/10.1002/hyp.10913, 2016

---

## Author Response (AR2)

**Editor Decision: Publish subject to minor revisions (further review by Editor)** (07 Jun 2017) by Jan Seibert

Comments to the Author:

Thanks for your efforts with the revision of the manuscript, which has greatly improved. As described nicely and in more detail by reviewer #3, there are still a number of issues which need to be addressed carefully. In particular, you need to better describe the M5 model (or models?) in a way that is understandable for the (hydrological) audience.

M5 model tree is the standard name thus, the plural will be M5 model trees. The text has been checked for coherence.

The explanation has been revised to clarify the process and to avoid duplicities in the used terms. Basically M5 hierarchically divides the input dataset in homogeneous parts such as, extreme precipitation and peak flows or summer low flows (which are likely to be characterized by periods without precipitation), and for each partition a linear model is adjusted. In the end it is a piece-wise linear model with each part dedicated to a particular hydrologic condition.

In Shortridge et al. (2016) this technique is simply described as follows: M5 models are a rule-based, non-parametric regression approach that fits a linear regression model to each terminal node of a regression tree (Quinlan, 1992). M5 models were fit using the Cubist package in R (Kuhn et al., 2014).

Certainly, a decision tree can be read as a set of rules. However such a definition neglects the fact that the dataset is divided in a hierarchical fashion, which has implications in the performance of decision trees (see Grubinger et al., 2014; Muñoz-Mas et al., 2016). Therefore, we advocated the original definition that involves the term 'decision tree'.

We hope the actual description is of sufficient clarity.

Grubinger, T., Zeileis, A., Pfeiffer, K.-P., 2014. Evtree: Evolutionary learning of globally optimal classification and regression trees in R. J. Stat. Softw. 61, 1–29.

Hettiarachchi, P., Hall, M.J., Minns, A.W., 2005. The extrapolation of artificial neural networks for the modelling of rainfall–runoff relationships. J. Hydroinformatics 7, 291–296.

Muñoz-Mas, R., Fukuda, S., Vezza, P., Martínez-Capel, F., 2016. Comparing four methods for decision-tree induction: A case study on the invasive Iberian gudgeon (Gobio lozanoi; Doadrio and Madeira, 2004). Ecol. Inform. 34, 22–34. doi:10.1016/j.ecoinf.2016.04.011

Shortridge, J.E., Guikema, S.D., Zaitchik, B.F., 2016. Machine learning methods for empirical streamflow simulation: A comparison of model accuracy, interpretability, and uncertainty in seasonal watersheds. Hydrol. Earth Syst. Sci. 20. doi:10.5194/hess-20-2611-2016

Solomatine, D.P., Dulal, K.N., 2003. Model trees as an alternative to neural networks in rainfall-runoff modelling. Hydrol. Sci. J. 48, 399–412. doi:10.1623/hysj.48.3.399.45291

Taghi Sattari, M., Pal, M., Apaydin, H., Ozturk, F., Sattari, M.T., Pal, M., Apaydin, H., Ozturk, F., 2013. M5 model tree application in daily river flow forecasting in Sohu Stream, Turkey. Water Resour. 40, 233–242. doi:10.1134/S0097807813030123

As the reference (Quinlan, 1992) is hard to access this description is really crucial. I would also like to see some better motivation on why this particular (type of) model was chosen. What are the advantages (disadvantages) compared to conceptual/bucket-type models (like SWAT, PRMS or HBV), which are otherwise often used in similar applications.

Process-based physical models are certainly the standard hydrological models (Shortridge et al. 2016). However, flexible data-driven machine learning techniques are gaining popularity because they can be based solely on precipitation and temperature (Shortridge et al. 2016) and can be automatized to perform multiple simulations. We selected M5 because it has shown to have skill in modelling daily streamflow (Solomatine and Dulal, 2003; Taghi Sattari et al., 2013), and is sufficiently fast to deal proficiently with larger datasets (Quinlan 2017). In addition, M5 is able to extrapolate while other approaches are not such as random forests (Shortridge et al., 2016) or multilayer perceptron (Hettiarachchi et al., 2005).

These comments have been included within the main text.

Please also get help from a native speaker, some terms just make little sense (future running flows, rear edge, ...), this is very confusing.

The term future running flows has been removed to avoid misunderstanding.

The "rear edge" concept is now better explained to make easier the understanding of the text.

Best regards,

Jan Seibert

Review comments for:

**Waning habitats due to climate change: the effects of changes in streamflow and temperature at the rear edge of the distribution of a cold-water fish**

**By:** José M. Santiago, Rafael Muñoz-Mas, Joaquín Solana, Diego García de Jalón, Carlos Alonso, Francisco Martínez-Capel, Javier Pórtoles, Robert Monjo, Jaime Ribalaygua

The subject paper represents a considerable investment in time and resources to assess the impact of climate change on the thermal habitat of a cold-water fish, the brown trout (Salmo trutta), in central Spain. It is an ambitious effort in collecting and organizing meteorologic, hydrologic and stream temperature data. The downscaling and biascorrection methods of the simulations from general circulation models are of satisfactory quality.

Where the work disappoints is in the methods used to develop state estimates of stream flow and water temperature. The artificial intelligence-based method, M5, used to simulate stream flows, is from a 27-year old paper, not readily available for study by reviewers. The description of the method (Pages 7-8) is replete with jargon and very difficult to understand. However, there is little reason to doubt that the authors applied the method incorrectly. In addition, it may well be the case that the results are within the bands of uncertainty that might be expected by applying a more modern hydrologic model. So, while the paper does not make a compelling case for using this method rather than a more modern one, the results are probably adequate for the specific scientific question posed here.

The stream temperature modeling is even less compelling. The regression-based methodology ( Eq. 1) is ad hoc and one that has been criticized for its lack of ability to project the effects of climate (Arismendi et al, 2014). The authors incorrectly cite the work of Piccolroaz et al, 2016 in support of their method. Rather than supporting regression methods like Eq. 1 of this paper, Piccolroaz et al, 2016, conclude that "Conversely, performances of purely regression-based or stochastic models are lower" than their model. It is a well-documented finding, however, that stream temperature is highly correlated with air temperature and, as is the case for the hydrologic model, M5, the results are likely to be within the uncertainty bands that would result from the application of one of the myriad models based on the thermal energy budget.

Arismendi *et al.* (2014) concluded that regression models based on air temperature can be inadequate for projecting future stream temperatures because they are only surrogates for air temperature, whereas Piccolroaz *et al.* (2016) argued that the adequacy depends on the hydrological regime, type of model and the time scale analysis. The main objections of these authors to regressive methods

arose when they modelled reaches of regulated rivers, but this is not our case. They recommend their models instead of the Mohseni model, but we have modified the last model to be adequate in a wider range of cases, giving information about the rivers by means of their parameters. So, we show that both the Mohseni model and, specially, our modified model implicitly integrates information on other factors, such as geology and flow regime. We have changed the wording of the phrase in the manuscript to avoid misunderstanding and clarify our arguments.

In two new columns in the Table 6, we include the values of two performance indicators: the Residual Standard Error (RSE) and the Nash Sutcliffe Efficiency coefficient (NSE). The high values of the performance indicators (excepting one case –Pirón 5) show that the models are sufficiently competent.

We are aware that the model can still be improved but important advances were made with respect to the models tested previously in the bibliography (Arismendi *et al*., 2014, Piccolroaz *et al*., 2016).

The development and analysis of this data set is noteworthy and worth publishing because of its environmental relevance. The outcomes from analyzing good, large data sets of stream temperature, hydrology and climate are reasonably robust in terms of the type of model being used. Based on this notion, it would seem the conclusions are also reasonable and, hopefully, of use to water resource planners.

The document also has some shortcomings in terms of an editorial nature, however, and would be improved in the following way:

* Have someone proofread it carefully. We are sensitive to the need to correctly transmit our work and for this reason, we have sent it to a prestigious service for review. So, the last version of this manuscript was revised by American Journals Experts (AJE) as shown by the attached certificate and invoice. We hope this will be good enough.

* There are too few statistical measures of outcomes, particularly for water temperature. We have included new performance indicators in Table 6. They are the residual standard error (RSE) and the Nash Sutcliffe Efficiency coefficient (NSE).

* Use a term other than "rear edge" to define the upper range of satisfactory temperatures for brown trout. The rear edge populations were defined by Hampe and Petit (2005) as those "*populations residing at the current low-latitude margins of species' distribution ranges, being disproportionately important for the long-term conservation of genetic diversity, phylogenetic history and evolutionary potential of species and that their investigation and conservation deserve high priority*". This is the eroding margin of the range where lineages mix, the genetic drift and local adaptations increase, and

droughts put populations under stress." *Rear edge populations are typically small and so isolated that regional population dynamics cannot easily compensate local extinction events*". We think that the use of the term "rear edge" improves the understanding of the ecological relevance of this zone. It is not the same concept as the upper limit of the thermal range. We clarify this definition in page 2, line 35.

* Explain "future running flows" or use a different term. The term future running flows has been removed to avoid misunderstanding.

* Define "agglomerative coefficient" The agglomerative coefficient (ac) is a measure of the clustering structure of the dataset, as expressed by Kaufman and Rousseeuw (2005). Its value ranges between 0 (maximum dissimilarity) and 1 (minimum dissimilarity). Thus, observed values in this study show high/reasonable structure.

$$ac = \frac{1}{n} \sum_{i=1}^{n} l(i)$$

* Try to use fewer acronyms. We believe that this is a difficult issue that may make reading tiresome. We include a list of symbols, abbreviations and acronyms.

* Table S1 is not referenced in the main document and needs a much better description of what's in it. Table is cited in page-12 line-14 of the previous version.

* Check the references to make certain they are complete. Checked.

References

Arismendi, I., Safeeq, M., Dunham, J.B. and Johnson, S.L.: Can air temperature be used to project influences of climate change on stream temperature? Environ. Res. Lett., 9(8), 084015. doi: 10.1088/1748-9326/9/8/084015, 2014.

Piccolroaz, S., Calamita, E., Majone, B., Gallice, A., Siviglia, A. and Toffolon, M.: Prediction of river water temperature: a comparison between a new family of hybrid models and statistical approaches: Prediction of River Water Temperature. Hydrol. Process., 30(21), 3901–3917. https://doi.org/10.1002/hyp.10913, 2016

**AMERICAN JOURNAL EXPERTS**

**EDITORIAL CERTIFICATE**

This document certifies that the manuscript listed below was edited for proper English language, grammar, punctuation, spelling, and overall style by one or more of the highly qualified native English speaking editors at American Journal Experts.

**Manuscript title:**

Waning habitats due to climate change: effects of streamflow and temperature changes at the rear edge of the distribution of a cold-water fish

**Authors:**

José M. Santiago, Rafael Muñoz-Mas, Joaquín Solana, Diego García de Jalón, Carlos Alonso, Francisco Martínez-Capel, Javier Pórtoles, Robert Monjo, Jaime Ribalaygua

**Date Issued:**

**Certificate Verification Key:**

628B-CF88-BF3C-D483-B53F

[Figure]

This certificate may be verified at www.aje.com/certificate. This document certifies that the manuscript listed above was edited for proper English language, grammar, punctuation, spelling, and overall style by one or more of the highly qualified native English speaking editors at American Journal Experts. Neither the research content nor the authors' intentions were altered in any way during the editing process. Documents receiving this certification should be English-ready for publication; however, the author has the ability to accept or reject our suggestions and changes. To verify the final AJE edited version, please visit our verification page. If you have any questions or concerns about this edited document, please contact American Journal Experts at support@aje.com.

American Journal Experts provides a range of editing, translation and manuscript services for researchers and publishers around the world. Our top-quality PhD editors are all native English speakers from America's top universities. Our editors come from nearly every research field and possess the highest qualifications to edit research manuscripts written by non-native English speakers. For more information about our company, services and partner discounts, please visit www.aje.com.

**American Journal Experts**

Send to:
Diego García de Jalón
Universidad Politécnica de Madrid
Oficina de Transferencia de Tecnologia
Ciudad Universitaria s/n
Madrid, Madrid
Spain, 28040
Q2818015F

American Journal Experts
601 West Main Street, Suite 102
Durham, NC 27701, United States
Phone: 1-919-704-4253
Fax: 1-919-287-2439
http://www.aje.com
Email: support@aje.com
Tax ID: 412141424

**Invoice**

Receipt code:  **LUHI-96A-0411141826**
Authors:  José M. Santiago, Rafael Muñoz-Mas, Joaquín Solana, Diego García de Jalón, Carlos Alonso, Francisco Martínez-Capel, Javier Pórtoles, Robert Monjo, Jaime Ribalaygua
Title:  Waning habitats due to climate change: effects of streamflow and temperature changes at the rear edge of the distribution of a cold-water fish
Submission date:  April 11 2017, 02:18 pm

| Invoice date | Description | Length | Time | Area of study | Price |
|---|---|---|---|---|---|
| April 11, 2017 | Standard Editing | Long (8001 - 10000 words) | 5 days | Hydrology | $348.00 |
| | | | | Re-edit discount | - $40.00 |
| | | | | PO surcharge | $25.00 |
| | | | | Remaining balance | $333.00 |

PAYMENT METHOD(S):
TERMS: Net 30 days. Online order
NOTES:
Bank transfer information:
Name on account: American Journal Experts
Account number: 1000079559885
Bank name and address: SunTrust Bank, 25 Park Place, Atlanta, GA 30303
Swift code: SNTRUS3A
ABA (branch): 061000104

**Your wire payment must reference your Receipt Code LUHI-96A-0411141826 for proper credit to your account.** In the case of overpayment through wire transfer, AJE will provide a credit to your account.

REMINDER: The $25.00 PO surcharge covers the AJE banking fees at the receiving end, but does NOT include the bank fee at your end. Please add the sending bank's fee to the total price of the invoice, unless otherwise noted on this invoice.

*Since we are based in the United States, we do not have an IBAN. Please use our bank account number and swift code for wire transfers. In place of VAT, please use our US Tax ID: 412141424